# Precise and Interpretable Editing of Code Knowledge in Large Language Models

**Min Xue**[1]**, Nikolai Bolik**[1]**, Lennart Stöpler**[1]**, Erik Imgrund**[2]**, Janik Schmid**[1]**,
Artur Andrzejak**[1]

[1]Institute of Computer Science, Heidelberg University, Germany
[2]BIFOLD & TU Berlin, Germany
{min.xue, bolik, lennart.stoepler, janik.schmid, artur.andrzejak}@uni-heidelberg.de
imgrund@tu-berlin.de

## Abstract

Large Language Models (LLMs) have demonstrated outstanding capabilities in various code-related tasks, including code completion, translation, or summarization. However, these pretrained models are static, posing a challenge to incorporate new knowledge into an LLM to correct erroneous behavior. Approaches such as retraining or fine-tuning demand extensive labeled datasets and might be computationally expensive, while prompt engineering fails to change models permanently. Knowledge Editing (KE) techniques (Wang et al., 2024) offer a more efficient alternative, enabling model updates with minimal data, even just a single example. Nevertheless, existing KE methods often manipulate parameters within the Transformer's multi-layer perceptrons (MLPs), where neuronal polysemanticity hinders both the precision and interpretability of the edits. To address these limitations, we exploit TransCoder (Dunefsky et al., 2024), an MLP-like model component with a wide and sparsely activated hidden feature vector. Specifically, we introduce **TransCoder-based Precise Editing** (TCPE), a novel method that leverages the sparsity and monosemanticity of the TransCoder's neurons for highly localized knowledge editing. TCPE exhibits neuron-level mechanistic interpretability characteristics, revealing the correspondence between the edited neurons and the specific code-related knowledge. Furthermore, we present **KECode**, a new evaluation benchmark for code-to-code translation based on functional equivalence (Wei et al., 2025). Using KECode, we conduct a systematic evaluation of representative KE methods in the context of code-to-code translation. Our experimental results demonstrate that TCPE outperforms existing KE methods, achieving a substantial improvement of translation accuracy of CodeLlama-7b-Instruct from 57.5% to 64.0% in a low-resource scenario of Java-to-D translation.

## 1 Introduction

Large Language Models (LLMs) have proved highly impactful in a multitude of fields within Software Engineering, including code summarization, code completion, code translation, software testing, program repair, and others (Hou et al., 2024; Jiang et al., 2024; Li et al., 2022; Sun et al., 2024). For code-related tasks, these models frequently need to be updated with new knowledge to correct erroneous behavior, accommodate changes in APIs or libraries, or align with developer preferences. However, this process is challenging due to the large volumes of training data required, high computational costs, and risks such as catastrophic forgetting, or loss of model consistency. Setting aside retraining or full-model finetuning (Li et al., 2024a; Zhu et al., 2024; GLM et al., 2024), even lightweight fine-tuning techniques such as LoRA still demand thousands of labeled training samples (Hu et al., 2022). Prompt engineering or external memory augmentation can provide superficial improvements but fail to fundamentally alter model behavior at the parameter level (Wang et al., 2025b; Wang & Zhu, 2024; Zhang et al., 2025).

In contrast, Knowledge Editing (KE) techniques (Wang et al., 2024) offer updating model knowledge with a small amount of data, typically single training examples, and promise precise model modifications, without impacting unrelated knowledge. To leverage this precision, we focus here on

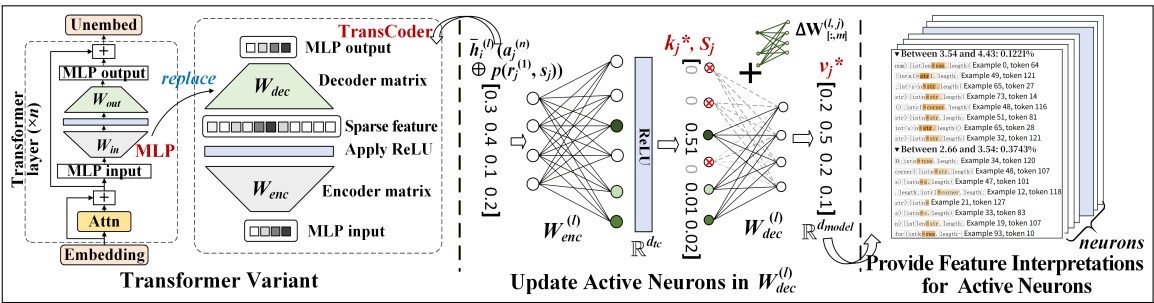

Figure 1: **Overview of TransCoder-Based Precise Editing.** The approach consists of: (1) Replacing the standard MLP module a TransCoder in a selected layer, yielding a Transformer variant, (2) An editing method which updates a minuscule fraction of TransCoder neurons relevant to the target knowledge, (3) Interpretation of neuron-level activations in TransCoder to reveal the link between the edited locations and the injected knowledge.

groundtruth-based local modification methods (classification from (Wang et al., 2024)), such as the popular approaches ROME (Meng et al., 2022) and MEMIT (Meng et al., 2023) proposed in context of Natural Language Processing (NLP).

ROME, MEMIT and related methods (e.g. PMET (Li et al., 2024b), FiNE (Pan et al., 2025)) perform updates within particular components of the Transformer architecture, the multi-layer perceptron (MLP) by conceptualizing it as a key-value store. However, these methods frequently face challenges in real-world scenarios due to model collapse (Yang et al., 2024a;b). They also might suffer from **limited specificity** and **poor interpretability**, which can be attributed to the polysemantic nature of MLP neurons (Scherlis et al., 2022).

To address the latter challenges, we propose a modified Transformer architecture that replaces the standard MLP layer with a TransCoder module (Dunefsky et al., 2024; Kissane et al., 2024a). A TransCoder is essentially a pair of an encoder and decoder matrices where the hidden feature vector is wider as in MLP and trained to be sparsely activated (e.g., via L1 regularization). Figure 1 gives an overview of our approach, which we refer to as **TransCoder-based Precise Editing** (TCPE). The key idea is to leverage the sparsity and monosemanticity of the TransCoder's activation space to automatically locate and edit neurons associated with target knowledge. This allows for precise updates while also enhancing interpretability, as the TransCoder's sparse activations can be directly linked to specific knowledge components.

Our second major contribution is **KECode**, a benchmark specifically designed for evaluating knowledge editing in context of code-to-code translation. Existing benchmarks for knowledge editing predominantly focus on on natural language-centric metrics such as efficacy, specificity, and reliability (Husein et al., 2025; Wang et al., 2024). These metrics may not be directly applicable or even meaningful in code-to-code translation, as here the primary success indicator is the functional equivalence (Glucksberg, 1984; Wei et al., 2025) of the original and translated code.

To bridge this gap, we propose a benchmark tailored for evaluating knowledge editing capabilities in the code domain (Chen et al., 2021) based on verifying functional equivalence. To this end we have collected a dataset of 600 Java-to-D code translation tasks. We have selected D as the target language due to its relative rarity, which allows us to create a low-resource setting for our experiments. Our benchmark comprises a translation step before knowledge edits, where a Transformer model is deployed to translate Java functions into corresponding D-language code. We then leverage unit tests provided for each example to check functional equivalence. Potential mistranslations (likely frequent due to the low-resource scenario) are clustered based on semantic similarity of the error messages (Islam & Inkpen, 2008). Subsequently, we inject for each error cluster the suitable correction knowledge into the model using the KE technique to be evaluated. Finally, the testing and clustering process is repeated to assess the effectiveness of the knowledge editing method according to multiple metrics described in Section 3.2. Overall, our contributions are as follows:

- We introduce TransCoder-based Precise Editing (TCPE), a neuron-level intervention method that leverages the sparsity and mono-semantic property of the TransCoder's activation space to precisely identify and update active neurons responsible for target knowledge.

- We develop KECode, a novel functional equivalence-based benchmark specifically designed to evaluate knowledge editing capabilities in a low-resource Java-D translation task.
- We demonstrate a neuron-level interpretability mechanism which effectively indicates the connection between the edited neurons and the inserted knowledge.
- We evaluate a collection of established knowledge editing methods, including ROME (Meng et al., 2022), MEMIT (Meng et al., 2023), PMET (Li et al., 2024b), AGRACE (Li et al., 2025), LoRA (Hu et al., 2022), among others, on the code-related task.
- We conduct extensive experiments which show that TCPE outperforms existing knowledge editing methods, significantly improving the translation accuracy of CodeLlama-7b-Instruct from 57.5% to 64.0% in a low-resource setting of Java-to-D translation.

The remainder paper is organized as follows. Section 2 introduces fundamental concepts and terms of the domain. We describe the approach in Section 3 and the experimental evaluation in Section 4. Lastly, we conclude this work in Section 5. Appendix comprises related work, additional experimental results, examples of interpretability experiments, and training details.

## 2 PRELIMINARIES

**Transformer and MLP Layer.** In general terms, an autoregressive Transformer model (Vaswani et al., 2017) can be described as a mapping $\mathcal{G} : \mathcal{X} \to \mathcal{Y}$ of an input sequence $x = [x_1, \ldots, x_T] \in \mathcal{X}$ over a vocabulary $V$ to a next-token probability distribution $y \in \mathcal{Y} \subset \Delta^{|V|}$. This mapping is operationalized as an iterative transformation of a hidden state $h$. First, the hidden state is initialized as the sum of a token embedding and possibly a positional embedding[1]: $h_i^{(0)} = emb(x_i) + pos(i) \in \mathbb{R}^{d_{\text{model}}}$, where $i$ denotes the token index and $d_{\text{model}}$ denotes the model dimensionality. Then the hidden state is passed through $L$ consecutive layers. At each layer $l \in 1, \ldots, L$, the hidden state $h_i^{(l)}$ for the $i$-th token is computed as:

$$h_i^{(l)} = \bar{h}_i^{(l)} + \text{MLP}^{(l)}(\bar{h}_i^{(l)}), \quad \bar{h}_i^{(l)} = h_i^{(l\text{-}1)} + \sum_{\text{head } n} \text{attn}^{(l,n)}(h_i^{(l-1)}; h_{1:i}^{(l-1)}), \tag{1}$$

where $\text{attn}^{(l,n)}(h_i^{(l\text{-}1)}; h_{1:i}^{(l\text{-}1)})$ denotes the output of the $n$-th attention head at layer $l$, given destination token $h_i^{(l\text{-}1)}$ and all preceding source tokens $h_{1:i}^{(l\text{-}1)}$. The function $\text{MLP}^{(l)}(\cdot)$ denotes the token-wise feed-forward transformation at layer $l$, defined as[2]:

$$\text{MLP}^{(l)}(\bar{h}_i^{(l)}) = \mathbf{W}_{\text{out}}^{(l)} \cdot \sigma\left(\mathbf{W}_{\text{in}}^{(l)} \cdot \gamma\left(\bar{h}_i^{(l)}\right)\right), \tag{2}$$

where $\mathbf{W}_{\text{in}}^{(l)} \in \mathbb{R}^{d_{\text{mlp}} \times d_{\text{model}}}$ and $\mathbf{W}_{\text{out}}^{(l)} \in \mathbb{R}^{d_{\text{model}} \times d_{\text{mlp}}}$ are the weight matrices of the two fully connected layers in the MLP. $\mathbf{W}_{\text{in}}^{(l)}$ transforms the input from the model's hidden dimension $d_{\text{model}}$ to the internal feature dimension $d_{\text{mlp}}$, and $\mathbf{W}_{\text{out}}^{(l)}$ projects it back to the original dimension $d_{\text{model}}$. Here, $\gamma(\cdot)$ denotes the layer normalization function, and $\sigma(\cdot)$ is the non-linear activation function. Finally, an unembedding matrix is applied and the resulting logits are projected onto the probability simplex using a softmax function.

**Transformer Variant and TransCoder Module.** As shown in Figure 1, TCPE extends the Transformer model $\mathcal{G}$ by replacing the MLP at layer $l^*$ with a TransCoder module. We call this modified model variant $\mathcal{A}$. Dunefsky et al. (2024) introduce TransCoder as a sparse approximation of the MLP layer:

$$z_{\text{TC}}^{(l)}(\bar{h}_i^{(l)}) = \text{ReLU}\left(\mathbf{W}_{\text{enc}}^{(l)} \cdot \bar{h}_i^{(l)}\right), \tag{3}$$

$$\text{TC}^{(l)}(\bar{h}_i^{(l)}) = \mathbf{W}_{\text{dec}}^{(l)} \cdot z_{\text{TC}}^{(l)}(\bar{h}_i^{(l)}), \tag{4}$$

where $\mathbf{W}_{\text{enc}}^{(l)} \in \mathbb{R}^{d_{\text{tc}} \times d_{\text{model}}}$ and $\mathbf{W}_{\text{dec}}^{(l)} \in \mathbb{R}^{d_{\text{model}} \times d_{\text{tc}}}$ are the encoder and decoder weight matrices. However, unlike traditional MLPs the TransCoder module is trained to minimize the approximation error alongside a sparsity loss (see Appendix L). Consequently, for a given input $\bar{h}$ only very few elements of $z_{\text{TC}}^{(l)}(\bar{h})$ are non-zero. We refer to these features as *active neurons*.

---

[1] Some variants (e.g. RoPE (Su et al., 2024)) place the positional embeddings inside the attention module.
[2] Throughout the paper all biases are omitted for brevity.

## 3 APPROACH

In this section, we introduce the TCPE method and describe the benchmark KECode.

### 3.1 TCPE: TRANSCODER-BASED PRECISE EDITING IN THE CODE DOMAIN

**Specifying Correction Knowledge in the Code Domain.** In code translation tasks, code LLMs map a source code snippet to a functionally equivalent target snippet (Galasso et al., 2022; Wei et al., 2025). To support knowledge editing applications, we represent each translation instance as a four-tuple $(r^{(1)}, s, r^{(2)}, o)$, where $s$ is the source code snippet (the *subject*) and $o$ is the functionally equivalent target snippet (the *object*). The prefix context $r^{(1)}$ includes the code preceding $s$, such as imports, comments, and prior definitions, and the suffix context $r^{(2)}$ contains the code following $s$ and may include the initial portion of $o$.

Specifically, we define the prompt as $p(r^{(1)}, s, r^{(2)}) = r^{(1)} \oplus s \oplus r^{(2)}$, where $\oplus$ denotes string concatenation (Meng et al., 2023). If the predicted $o$ contains syntax errors or violates functional equivalence, we manually correct it to $o^*$. The resulting tuple $(r^{(1)}, s, r^{(2)}, o^*)$ is referred to as *correction knowledge*. An example is provided in Appendix I. (Complementarily, building on the causal intervention method (Meng et al., 2022), we introduce a *fine-grained causal intervention method* to further examine the role of the subject $s$ in the four-tuple $(r^{(1)}, s, r^{(2)}, o)$ across different programming languages (see Appendix H).)

**Neuron-Level Sparse Update Mechanism.** Building on ROME (Meng et al., 2022), we model the TransCoder decoder weight $\mathbf{W}_{\text{dec}}^{(l)} \in \mathbb{R}^{d_{\text{model}} \times d_{\text{tc}}}$ as a linear associative memory that maps a key $k \in \mathbb{R}^{d_{\text{tc}}}$ to a value $v \in \mathbb{R}^{d_{\text{model}}}$ via $\mathbf{W}_{\text{dec}}^{(l)} k = v$. To precisely inject correction knowledge, we propose to only target the active neurons in the Transformer variant $\mathcal{A}$. Given $T$ new key-value pairs

$$\{(k_j^*, v_j^*, S_j)\}_{j=1}^T, \quad k_j^* \in \mathbb{R}^{d_{\text{tc}}}, \ v_j^* \in \mathbb{R}^{d_{\text{model}}} \ ,$$

where $(k_j^*, v_j^*)$ encodes the $j$-th correction knowledge $(r_j^{(1)}, s_j, r_j^{(2)}, o_j^*)$. The set $S_j = \{a \in [d_{\text{tc}}] \mid (k_j^*)_a > \tau\}$ contains the indices of activation values in $k_j^*$ that exceed the threshold $\tau$, where $(k_j^*)_a$ denotes the activation at position $a$. ( Appendix G analyzes the overlap of $S_j$ across different error types under both MLP and TransCoder modules, where TransCoder exhibits low cross-error overlap, indicating specialized neuron activation for distinct error types.)

Following ROME (Meng et al., 2022), for each key-value pair $(k_j^*, v_j^*, S_j)$, we compute the update matrix $\Delta \mathbf{W}^{(l,j)} = \frac{(v_j^* - \mathbf{W}_{\text{dec}}^{(l)} k_j^*)}{(C^{-1} k_j^*)^\top k_j^*} \cdot (C^{-1} k_j^*)^\top$. We estimate the covariance matrix $C \in \mathbb{R}^{d_{\text{tc}} \times d_{\text{tc}}}$ using samples from the "bigcode/the-stack[3]" dataset. Different to standard ROME, we restrict updates to the active neurons indexed by $S_j$, enabling precise modifications at the neuron level:

$$\mathbf{W}_{\text{dec}}^{(l)'}[:, m] = \mathbf{W}_{\text{dec}}^{(l)}[:, m] + \Delta \mathbf{W}^{(l,j)}[:, m], \quad \forall m \in S_j \ , \tag{5}$$

where each $\Delta \mathbf{W}^{(l,j)}$ is a rank-one update matrix, sparsified via the active neuron index set $S_j$, ensuring that only relevant neurons are updated, thereby enhancing specificity and minimizing interference with unrelated knowledge.

**Encoding Correction Knowledge.** Unlike knowledge editing in natural language (Meng et al., 2022; 2023), the generation of $(k_j^*, v_j^*)$ in the code domain relies on the knowledge four-tuple $(r^{(1)}, s, r^{(2)}, o^*)$. For each error type, we encode the correction knowledge $(r_j^{(1)}, s_j, r_j^{(2)}, o_j^*)$ into a key-value pair $(k_j^*, v_j^*)$ through the following two steps.

*Step 1: Generating $k_j^*$.* We define $k_j^*$ as the mean post-activation output from the TransCoder encoder at the final token position $i$ of the prompt $p(r_j^{(1)}, s_j)$. Specifically, we construct $N$ input sequences by prepending randomly sampled prefixes $\{a_j^n\}_{n=1}^N$ to the prompt $p(r_j^{(1)}, s_j)$, where $p(r_j^{(1)}, s_j) = r_j^{(1)} \oplus s_j$. For each composite input $a_j^n \oplus p(r_j^{(1)}, s_j)$, we process it through Transformer

---

[3]https://huggingface.co/datasets/bigcode/the-stack-v2-dedup

architecture $\mathcal{A}$, and extract the non-linear activation $z_{\text{TC}}^{(l)}(\cdot)$ from TransCoder encoder at the final token position $i$ of $p(r_j^{(1)}, s_j)$. Finally, we compute $k_j^*$ as the average activation across all $N$ sequences. Formally, the $k_j^*$ is computed as:

$$k_j^* = \frac{1}{N} \sum_{n=1}^{N} z_{\text{TC}}^{(l)} \left( \bar{h}_i^{(l)}(a_j^n \oplus p(r_j^{(1)}, s_j)) \right). \tag{6}$$

Here, $\bar{h}_i^{(l)}(a_j^n \oplus p(r_j^{(1)}, s_j))$ denotes the attention output (with residual) at layer $l$ for the final token of the input sequence $a_j^n \oplus p(r_j^1, s_j)$.

*Step 2: Generating $v_j^*$.* We seek to construct a value vector $v_j^*$ that encodes the new relation $(r_j^{(2)}, o_j^*)$ as an attribute of the prompt $(r_j^{(1)}, s_j)$. To implement this, we introduce a minimal perturbation $\delta\_j$ to the TransCoder's output. Specifically, the perturbation $\delta_j$ is added to the output of the TransCoder decoder $\mathbf{W}_{\text{dec}}^{(l)}$, at the final token position of the input sequence $a_j^n \oplus p(r_j^{(1)}, s_j, r_j^{(2)})$, guiding the model to predict the new target object $o_j^*$. Formally, this process is expressed as:

$$v_j^* = TC^{(l)} + \arg\min_{\delta_j} \left( \frac{1}{N} \sum_{n=1}^{N} -\log P_{\mathcal{A}(TC^{(l)}+=\delta_j)}[o_j^* \mid a_j^n \oplus p(r_j^1, s_j, r_j^2)] \right). \tag{7}$$

where $\mathcal{A}(TC^{(l)} + \delta_j)$ denotes the addition of the perturbation $\delta_j$ to the TransCoder output $TC^{(l)}$ within the Transformer architecture $\mathcal{A}$. Once the corrected knowledge $\{(r_j^{(1)}, s_j, r_j^{(2)}, o_j^*)\}_{j=1}^{T}$ is encoded as $\{(k_j^*, v_j^*, S_j)\}_{j=1}^{T}$, we apply Equation (5) to selectively update the active neurons in the TransCoder decoder layer $\mathbf{W}_{\text{dec}}^{(l)}$.

## 3.2 KECode: Knowledge Editing Benchmark in Low-Resource Code Translation

Unlike natural language, programming languages require strict syntactic and semantic correctness. Even with successful knowledge injection, generated code may still fail to compile or exhibit functional errors. Therefore, in the code domain, knowledge editing should be evaluated based on functional equivalence rather than superficial textual similarity (Wei et al., 2025).

**Dataset Construction: G4GD.** Following the principle of functional equivalence, we construct the G4GD dataset for the low-resource Java-to-D translation task. We adopt the GeeksforGeeks[4] dataset provided by CodeGen as our foundation, which contains hundreds of Java functions. To support automated evaluation, we collect 10 representative input-output pairs from each Java function and develop corresponding unit tests in the D language. The final G4GD dataset comprises 600 Java functions, each paired with 10 independent D unit tests.

In Appendix J.2, we provide a detailed comparison between the G4GD dataset and widely used benchmarks such as HumanEval (Chen et al., 2021) and MBPP (Austin et al., 2021). Appendix J.6 includes the prompt formulation that guides the model in translating the source code into functionally equivalent functions in the target language. The G4GD dataset is publicly available at Hugging Face[5].

**Functional Error Clustering Mechanism.** We categorize the generated D functions based on the textual similarity of the compilation messages. Specifically, in the Java-to-D translation task, we use each Java function $x_s$ from the G4GD dataset to generate a corresponding D function $y_s$ via a Code LLM, yielding 600 translation pairs $(x_s, y_s)$, $s \in [1, 600]$. We then execute unit tests to assess the correctness of each generated D function $y_s$, and collect the compilation messages, including runtime error messages (if compilation fails) or success indicators (if compilation succeeds). For failed cases, we extract the first six tokens of the error message, denoted as $g_s$. Each message is paired with its corresponding translation pair $(x_s, y_s)$, resulting in a dataset $\mathcal{D}_{\text{full}} = \{(x_s, y_s, g_s)\}_{s=1}^{600}$.

Based on the compilation logs $g_s$, we partition $\mathcal{D}_{\text{full}}$ into three subsets: $C_{\text{succ}}$ (compiles and passes all tests), $C_{\text{FailPass}}$ (compiles but fails some tests), and $C_{\text{incomp}}$ (fails to compile). Then, we encode the error messages $g_s$ using the "gte-base-en-v1.5[6]" model and cluster them via cosine similarity

---

[4]https://github.com/yakuhzi/c2c-translation/tree/main/data
[5]https://huggingface.co/datasets/AIP-Heidelberg/G4GD
[6]https://huggingface.co/Alibaba-NLP/gte-base-en-v1.5

(threshold 0.9), yielding $A$ error clusters $C_i \subseteq C_{\text{incomp}}, i \in [1, A]$. (Appendix J provides detailed error cluster statistics, intra-cluster examples, and the error message list for CodeLlama-7b-Instruct.)

**Evaluation Protocol.** To evaluate the performance of knowledge editing in the code domain, we designed a functional equivalence-based evaluation framework focusing on three key metrics: *Efficacy*, *Specificity*, and *Reliability*. In particular, using Java functions from the G4GD dataset as inputs, we construct pre-edit and post-edit datasets, $\mathcal{D}_{\text{full}} = \{(x_s, y_s, g_s)\}_{s=1}^{600}$ and $\mathcal{D}'_{\text{full}} = \{(x_s, y'_s, g'_s)\}_{s=1}^{600}$. The pre-edit dataset $\mathcal{D}_{\text{full}}$ is partitioned into three subsets: $C_{\text{succ}}$, $C_{\text{FailPass}}$, and a collection of error clusters $C_{\text{incomp}} = \{C_i\}_{i=1}^{A}$. Similarly, the post-edit dataset $\mathcal{D}'_{\text{full}}$ is divided into $C'_{\text{succ}}$, $C'_{\text{FailPass}}$, and $C'_{\text{incomp}} = \{C'_j\}_{j=1}^{B}$, where $A$ and $B$ represent the numbers of distinct error clusters before and after editing, respectively. If $i = j$, then $C_i$ and $C'_j$ correspond to the same error type. Furthermore, based on the target error type(s) of the edit, we define their union as the *pre-edit target group* $C_{\text{target}}$, and its complement within $\mathcal{D}_{\text{full}}$ as the *pre-edit non-target group*, $C_{\text{non-target}} = \mathcal{D}_{\text{full}} \setminus C_{\text{target}}$. The *post-edit target group*[7] $C'_{\text{target}}$ and *post-edit non-target group* $C'_{\text{non-target}}$ correspond to the same error types as $C_{\text{target}}$ and $C_{\text{non-target}}$, respectively. Based on this, we define the following evaluation metrics.

*Efficacy:* It measures how effectively the edit corrects targeted errors and comprises two metrics: (1) *Generalization* (GN), defined as the proportion of samples in the pre-edit target group $C_{\text{target}}$ that are correctly translated post-editing: $\mathbb{E}_{x_s \sim \pi_X(C_{\text{target}})} \mathbb{I}\{(x_s, y'_s, g'_s) \in C'_{\text{succ}}\}$, where $\pi_X(\cdot)$ is the projection operator (Codd, 1970) extracting $x_s$ from triplets $(x_s, y'_s, g'_s)$, and $\mathbb{I}(\cdot)$ is the indicator function. (2) *Cluster Drift* (CD), which measures the relative change in the cardinality of the target error group after editing, computed as $|C'_{\text{target}}|/|C_{\text{target}}|$.

*Specificity:* It quantifies the extent to which the edit avoids unintended changes and is measured by two complementary metrics: (1) *Locality* (LoC), which evaluates the consistency of error categories within $C_{\text{non-target}}$ pre- and post-edit: $\mathbb{E}_{x_s \sim \pi_X(C_{\text{non-target}})} \mathbb{I}\{(x_s, y'_s, g'_s) \in C'_{\text{non-target}}\}$. (2) *Destructiveness* (DT) is defined as the proportion of originally correct samples that become incorrect after editing: $\mathbb{E}_{x_s \sim \pi_X(C_{\text{succ}})} \mathbb{I}\{(x_s, y'_s, g'_s) \notin C'_{\text{succ}}\}$.

*Reliability (RE):* It measures the global impact of the edit on the model's overall accuracy, defined as the ratio of post-edit accuracy ($\text{AC}_{post}$) to the pre-edit accuracy ($\text{AC}_{pre}$): $\mathbb{E}_{x_s \sim \pi_X(\mathcal{D}'_{\text{full}})} \mathbb{I}\{(x_s, y'_s, g'_s) \in C'_{\text{succ}}\} / \mathbb{E}_{x_s \sim \pi_X(\mathcal{D}_{\text{full}})} \mathbb{I}\{(x_s, y_s, g_s) \in C_{\text{succ}}\}$.

## 4 EXPERIMENTS

In our study, we (i) investigate TCPE's interpretability (Section 4.2 and Appendix F), (ii) extend mainstream knowledge editing methods to the code domain for comparative evaluation (Section 4.3 and Appendix E.1), (iii) examine the information-carrying capacity of active neurons to substantiate precise editing (Section 4.3 and Appendix E.2), and (iv) broaden our study to general NLP tasks to further probe the origins of low specificity in ROME-based approaches (Appendix E.3). Furthermore, we analyze the effects of TransCoder size and layer positions, as well as overlaps of active neurons, with detailed results provided in Section 4.3 and Appendix G. (The code is available at the following link[8].)

### 4.1 EXPERIMENTAL DETAILS

**Base Models and TransCoder Variants.** In our work, we adopt CodeLlama-7b-Instruct and Llama-2-7b (hereafter CodeLlama and Llama2) as base models, and construct a series of Transformer variants with varying TransCoder widths for systematic analysis. We utilize the "TransformerLens[9]" framework to build four CodeLlama variants with varying TransCoder intermediate dimensions ($d_{\text{tc}}$): $\text{LTC}_{mlp}$ ($d_{\text{tc}} = d_{\text{mlp}} = 11,008$), LTC4 ($d_{\text{tc}} = 4,096 * 4$), LTC8 ($d_{\text{tc}} = 4,096 * 8$), and LTC16 ($d_{\text{tc}} = 4,096 * 16$), as well as two Llama2 variants: MTC4 ($d_{\text{tc}} = 4,096 * 4$) and MTC8 ($d_{\text{tc}} = 4,096 * 8$). Here, each variant is constructed by replacing a single MLP layer at $l \in \{10, 19, 23\}$ with the TransCoder. Notably, we designed the TransCoder Adapter to enable fast integration of TransCoder for the above variants in just a few seconds. (See Appendices L.3 and L.2 for details.)

---

[7]Knowledge editing may cause certain target error types to be absent in the post-edit (non-)target group. This is considered in our evaluation design and does not affect metric validity.

[8]https://github.com/minxue29031/TCPE

[9]https://github.com/TransformerLensOrg/TransformerLens

**Datasets and Evaluation Metrics.** We evaluate knowledge editing performance on the **KECode** benchmark (including the G4GD dataset), **HumanEval** (Chen et al., 2021), **zsRE** (Levy et al., 2017), and **CounterFact** (Meng et al., 2022). For the G4GD dataset, we observe that the dominant error types remain consistent across CodeLlama, LTC4, and LTC8, with clusters $C_0$, $C_8$, $C_4$, and $C_6$ together accounting for $57.68\% \sim 58.51\%$ of $C_{\text{incomp}}$. Accordingly, our editing evaluation focuses on these clusters, assessing performance in both single-error and multi-error scenarios, using a functional equivalence-based framework with three key metrics: *Efficacy*, *Specificity*, and *Reliability*. For HumanEval, we leverage it to examine the broader impact of knowledge injection after G4GD-based edits, using the *Reliability* metric to measure overall model performance (see Section 3.2). For CounterFact and zsRE, we follow the experimental protocols of Pan et al. (2025), focusing on three key metrics: *Efficacy*, *Specificity*, and *Generalization* (see Appendix D for details).

**Baselines.** Meng et al. (2022) demonstrate that factual knowledge is primarily stored in the middle MLP layers of Transformers. Using a fine-grained causal intervention scanning all layers and token positions, we confirm that code knowledge is similarly localized in these middle layers (see Appendix H). Furthermore, we observe that TransCoder modules with the same width exhibit consistent sparsity patterns (i.e., the average number of activated neurons per token) and achieve comparable performance when replacing MLP layers at different positions (see Section 4.3 and Figures 10(c) and 10(f) in the Appendix). Accordingly, our subsequent TCPE editing experiments primarily target the middle TransCoder modules at layer 19.

To assess the effectiveness of TCPE, we compare it against representative knowledge editing baselines: ROME (Meng et al., 2022), MEMIT (Meng et al., 2023), PMET (Li et al., 2024b), FiNE (Pan et al., 2025), Fine-Tuning (FT) (Zhu et al., 2020b), AGRACE (Li et al., 2025), Few-shot (Parnami & Lee, 2022), WISE (Wang et al., 2025a), and LoRA (Hu et al., 2022). (Method descriptions and hyperparameters for TCPE and baselines are provided in Appendices D.3 and K.)

## 4.2 INTERPRETABILITY ASSESSMENT OF TCPE

In this section, we first use TCPE to analyze the relationship between injected knowledge and neurons with varying activation levels, and then compare the interpretability of TransCoder and MLP neurons.

**Neuron-Level Interpretability in Knowledge Editing.** We employ TCPE on LTC4 within the G4GD dataset to explore whether active features during knowledge editing align with specific error types, providing insights into neuron-level interpretability. Focusing on a typical D-type conversion error "*Error: cannot implicitly convert expression 'str.length' of type 'ulong' to 'int'*", we analyze the interpretability differences between active and inactive features in the activation $k_j^*$ from the LTC4 TransCoder module. Notably, only 57 features are active, representing 0.348% of the intermediate dimension $d_{\text{tc}}$. Specifically, we first record the indices of the top-10 most active features and 10 randomly selected inactive features in $k_j^*$ during the injection of correction knowledge for this error type. Then, using all Java samples and D samples in $C_8$ as input, we capture and analyze the patterns of the top-activating examples for both the top-10 active features and 10 inactive features. Figure 2 shows a representative example from the top-10 active features, which consistently respond to key tokens such as 'str', 'string', or '=.length', directly related to the target error. (More results can be found in Appendix F.2). In contrast, Appendix F.3 presents examples from 10 randomly inactivated features, which respond to structural or control-flow tokens like 'if', 'N', 'ps', and ';'. Although these inactive features

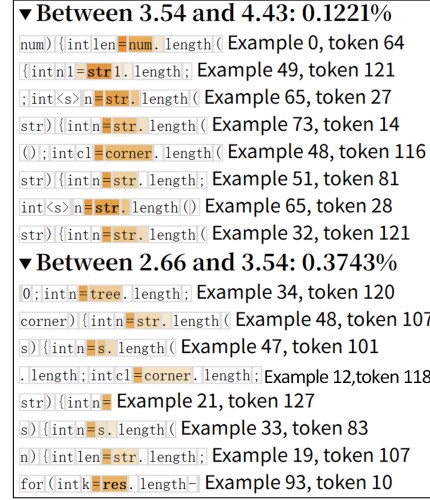

Figure 2: **Top-Activating Examples for Active Neurons.**

also exhibit stable activation patterns, their captured knowledge is largely unrelated to the target error. This comparison highlights that the highly active features exhibit semantic specificity and show direct correlation with the target error, providing interpretability for knowledge editing.

**Blind Interpretability Comparison of TransCoders and MLPs.** Following the methodology of Dunefsky et al. (2024), we evaluate the interpretability of TransCoder features compared to the MLP

features. Here, a feature is considered interpretable if it exhibits clear and consistent patterns (e.g., syntactic or semantic) across the input examples that activate it (Kissane et al., 2024b; Bloom, 2024). We randomly selected 50 features from the TransCoder module ($\mathbf{W}_{enc}^{(19)}$) of LTC4 and 50 features from the MLP layer ($\mathbf{W}_{in}^{(19)}$) of CodeLlama. For each feature, we precomputed the *top-activating examples* from a pool of 37,055 Java and D code samples sourced from the "semeru/code-text-java[10]" and "UKPLab/SLTrans[11]" corpora. To avoid bias, features were randomly shuffled prior to analysis. Then, we performed a blind manual evaluation to determine whether the top-activating examples exhibited interpretable patterns, categorized as "*uninterpretable*", "*possibly-interpretable*", or "*interpretable*" (including

Table 1: **Interpretability Analysis of MLP and TransCoder (LTC4) Features.**

| Type | TransCoder | MLP |
|---|---|---|
| Interpretable | 33 | 4 |
| Possibly-Interpretable | 8 | 5 |
| Uninterpretable | 9 | 41 |

a subset labeled "*context-free*", triggered by individual tokens). Examples of each category are shown in Appendix F.1. After all annotations were completed, the feature source (TransCoder or MLP) was revealed. As shown in Table 1, features from the TransCoder module demonstrate higher interpretability compared to those from the MLP layer, aligned with prior findings on model interpretability by Dunefsky et al. (2024). Furthermore, in Appendix G, we evaluate the overlap of high-activation neurons across distinct error clusters, revealing that TransCoder neurons exhibit more independent activation patterns than the MLP, thereby providing further support for the above results.

Table 2: **Comparative Performance of Knowledge Editing Methods in a Multi-Error Editing Scenario.** *Score* quantifies overall knowledge editing performance and is calculated as Score = $(\text{GN} - \text{CD}) + (\text{LoC} - \text{DT}) + \text{RE}$.

| Method | Score↑ | Efficacy | | Specificity | | Reliability | | |
|---|---|---|---|---|---|---|---|---|
| | | GN↑ | CD↓ | LoC↑ | DT↓ | RE↑ | $AC_{post}$ | $AC_{pre}$ |
| FT | 86.40 | 7.09 | 100.00 | 87.36 | 6.38 | 97.97 | 56.33 | 57.50 |
| Few-shot | 120.20 | 12.06 | 87.23 | 90.20 | 2.61 | 107.78 | 61.33 | 57.50 |
| WISE | 99.74 | 3.55 | 98.58 | 96.51 | 1.74 | 100.00 | 57.50 | 57.50 |
| AGRACE | 99.85 | 0.71 | 100.71 | 99.56 | 0.00 | 100.29 | 57.67 | 57.50 |
| FiNE | 144.50 | 29.08 | 54.61 | 81.92 | 13.62 | 101.74 | 58.50 | 57.50 |
| LoRA | 147.27 | 29.08 | 58.87 | 82.57 | 10.44 | 104.93 | 60.33 | 57.50 |
| ROME | 109.60 | 29.79 | 80.14 | 76.47 | 16.23 | 99.71 | 57.33 | 57.50 |
| MEMIT | 14.96 | 7.09 | 124.82 | 72.99 | 22.32 | 82.03 | 47.17 | 57.50 |
| PMET | -4.70 | 12.77 | 104.26 | 55.77 | 38.55 | 69.57 | 40.00 | 57.50 |
| TCPE ($LTC_{mlp}$) | 171.59 | 30.61 | **44.90** | 83.00 | 6.05 | 108.93 | 63.00 | 57.83 |
| TCPE (LTC4) | 171.45 | **32.86** | 46.43 | 82.17 | **6.86** | 109.71 | **64.00** | 58.33 |
| TCPE (LTC8) | **174.82** | 31.66 | 49.64 | **88.50** | 5.73 | **110.03** | **64.00** | 58.17 |

## 4.3 EDITING PERFORMANCE OF TCPE

**Analysis of TCPE and Baselines in a Multi-Error Editing Scenario.** Table 2 presents a comparison of TCPE and baseline methods in the multi-error editing scenario on the G4GD dataset. TCPE outperforms all baselines across key metrics, including efficacy, specificity, and reliability. Due to the inherently strict syntactic and semantic constraints of programming languages, knowledge editing in the code domain is highly sensitive to the granularity of interventions. Compared to broader interventions like MEMIT and PMET, single-layer approaches (e.g., ROME) tend to yield more reliable outcomes. In this work, TCPE builds upon these approaches by combining TransCoder's sparse representations with precise neuron-level interventions. This method enables strong generalization while minimizing unintended side effects, as reflected in its high specificity (in terms of locality and destructiveness). Crucially, TCPE surpasses baselines in reliability, achieving significant performance gains through effective knowledge insertion. It demonstrating TCPE's ability to perform effective knowledge edits while preserving model integrity.

**Information-Carrying Role of Active Neurons in Supporting Precise Editing.** This study investigates the information-carrying roles of highly and lowly active neurons during knowledge injection to evaluate the feasibility of precise model editing. We apply TCPE to conduct conventional experiments on high-activation neurons and perform ablation studies on low-activation neurons on G4GD dataset.

---

[10]https://huggingface.co/datasets/semeru/code-text-java
[11]https://huggingface.co/datasets/UKPLab/SLTrans

Table 3: **Role of Active TransCoder Neurons in Multi-Error Editing.** Neurons are updated if activations $acv$ exceed thresholds $\tau \in \{0, 0.001, 0.01, 0.05, 0.08, 0.1, 0.15, 0.2\}$. "$\cup$" and "$\cap$" denote the union and intersection of updated neurons across multiple error types.

| Method | Efficacy | | Specificity | | Reliability | | | Neurons | |
|---|---|---|---|---|---|---|---|---|---|
| | GN↑ | CD↓ | LoC↑ | DT↓ | RE↑ | $AC_{post}$ | $AC_{pre}$ | $\cup$ | $\cap$ |
| LTC4 | | | | | | | | | |
| $acv$>0.2 | 24.29 | 59.29 | 82.39 | 7.43 | 105.43 | 61.50 | 58.33 | 34 | 3 |
| $acv$>0.15 | 27.86 | 61.43 | 82.61 | 7.14 | 107.14 | 62.50 | 58.33 | 42 | 3 |
| $acv$>0.1 | 32.86 | 46.43 | 82.17 | 6.86 | 109.71 | 64.00 | 58.33 | 69 | 3 |
| $acv$>0.08 | 34.29 | 52.86 | 81.96 | 9.43 | 106.57 | 62.17 | 58.33 | 78 | 4 |
| $acv$>0.05 | 35.00 | 48.57 | 82.39 | 8.57 | 106.86 | 62.33 | 58.33 | 98 | 4 |
| $acv$>0.01 | 33.57 | 50.00 | 83.04 | 8.29 | 106.29 | 62.00 | 58.33 | 135 | 4 |
| $acv$>0.001 | 32.86 | 50.71 | 83.04 | 8.29 | 106.00 | 61.83 | 58.33 | 147 | 5 |
| $acv$>0 | 32.86 | 50.71 | 83.04 | 8.29 | 106.00 | 61.83 | 58.33 | 147 | 5 |
| LTC8 | | | | | | | | | |
| $acv$>0.2 | 29.50 | 51.80 | 88.29 | 6.59 | 108.31 | 63.00 | 58.17 | 39 | 1 |
| $acv$>0.15 | 30.22 | 49.64 | 87.85 | 6.59 | 108.88 | 63.33 | 58.17 | 51 | 2 |
| $acv$>0.1 | 32.37 | 46.76 | 87.42 | 6.59 | 109.74 | 63.83 | 58.17 | 70 | 3 |
| $acv$>0.08 | 31.66 | 49.64 | 88.50 | 5.73 | 110.03 | 64.00 | 58.17 | 92 | 3 |
| $acv$>0.05 | 32.37 | 51.08 | 82.86 | 9.74 | 106.59 | 62.00 | 58.17 | 122 | 4 |
| $acv$>0.01 | 31.66 | 51.08 | 82.65 | 10.32 | 105.73 | 61.50 | 58.17 | 181 | 5 |
| $acv$>0.001 | 31.66 | 51.08 | 82.65 | 10.32 | 105.73 | 61.50 | 58.17 | 195 | 5 |
| $acv$>0 | 31.66 | 51.08 | 82.65 | 10.32 | 105.73 | 61.50 | 58.17 | 197 | 5 |

Table 4: **Ablation on Active TransCoder Neurons in Multi-Error Editing.** Neurons are updated if activations $acv$ below thresholds $\tau \in \{0, 0.001, 0.01, 0.05, 0.08, 0.1\}$. Here, "Updated Neurons" denotes the number of neurons that meet each threshold.

| Method | Efficacy | | Specificity | | Reliability | | | Updated Neurons |
|---|---|---|---|---|---|---|---|---|
| | GN↑ | CD↓ | LoC↑ | DT↓ | RE↑ | $AC_{post}$ | $AC_{pre}$ | |
| LTC4 | | | | | | | | |
| $acv$ =0 | 1.43 | 101.43 | 98.48 | 0.57 | 100.29 | 58.5 | 58.33 | 16,237 |
| $acv$ ≤0.001 | 1.43 | 101.43 | 98.48 | 0.57 | 100.29 | 58.5 | 58.33 | 16,237 |
| $acv$ ≤0.01 | 1.43 | 101.43 | 98.26 | 0.57 | 100.29 | 58.5 | 58.33 | 16,249 |
| $acv$ ≤0.05 | 1.43 | 101.43 | 98.04 | 0.86 | 100 | 58.33 | 58.33 | 16,286 |
| $acv$ ≤0.08 | 1.43 | 101.43 | 98.04 | 0.86 | 100 | 58.33 | 58.33 | 16,306 |
| $acv$ ≤0.1 | 7.14 | 92.86 | 96.09 | 1.71 | 102 | 59.5 | 58.33 | 16,315 |
| LTC8 | | | | | | | | |
| $acv$ =0 | 1.44 | 100 | 99.13 | 0.29 | 100.57 | 58.5 | 58.17 | 32,571 |
| $acv$ ≤0.001 | 1.44 | 100 | 99.13 | 0.29 | 100.57 | 58.5 | 58.17 | 32,573 |
| $acv$ ≤0.01 | 1.44 | 100 | 99.13 | 0.29 | 100.57 | 58.5 | 58.17 | 32,587 |
| $acv$ ≤0.05 | 1.44 | 100 | 99.13 | 0.29 | 100.57 | 58.5 | 58.17 | 32,646 |
| $acv$ ≤0.08 | 1.44 | 99.28 | 99.13 | 0.29 | 100.86 | 58.67 | 58.17 | 32,676 |
| $acv$ ≤0.1 | 1.44 | 99.28 | 98.92 | 0.57 | 100.57 | 58.5 | 58.17 | 32,698 |

(1) *Active TransCoder Neurons.* In the multi-error editing scenario, less than 1% of neurons in the TransCoder decoder are active. To further investigate the information-carrying capacity of active neurons during knowledge editing, we experimented with different activity thresholds $\tau \in \{0, 0.001, 0.01, 0.05, 0.08, 0.1, 0.15, 0.2\}$. As shown in Table 3, the overall score remains relatively stable across a range of activation thresholds $\tau$, demonstrating the robustness of active neurons in supporting knowledge editing. Notably, even when only a small subset of highly active neurons ($acv > 0.2$) is updated, the performance remains competitive. This suggests that highly active neurons tend to carry more essential information during knowledge injection. Moreover, as the threshold is relaxed, the number of neurons participating in the update ($\cup$) increases, while the intersection of activated neurons across different error types ($\cap$) remains small. This highlights the specificity of TransCoder's neurons, as each neuron tends to respond to a specific error type, with minimal overlap between neurons involved in correcting different errors.

(2) *Ablation Study of Active TransCoder Neurons.* Based on the ablation experiments presented in Table 4, we provide a more comprehensive analysis of the relationship between neuron activation levels and the effectiveness of knowledge injection. As the activation threshold increases from 0 to 0.1, a larger number of low-activation and inactive neurons are included in the update process. Despite broadening the update scope, we observe no tangible gains in generalization. In LTC4, the number of updated neurons increases from 16,237 to 16,306 while the score remains 1.43. In LTC8, the updated neuron count reaches 32,698, yet the score remains unchanged at 1.44. Meanwhile, although specificity metrics (including locality and destructiveness) and reliability remain numerically high, this outcome primarily results from the model's inability to execute effective edits. Even when all low-activation and inactive neurons are updated, the model still fails to perform meaningful knowledge injection. These results suggest that, in the process of knowledge injection, a few high-activation neurons carry most of the relevant information, whereas the vast majority of low-activation neurons contribute little, providing support for precise knowledge editing.

**Effect of TransCoder Size and Layer Position.** We evaluate the functional fidelity of TransCoder by examining whether replacing the MLP layer with a TransCoder module across different layers affects model performance. As shown in Table 5, model performance remains stable across different TransCoder widths and positions on the G4GD dataset. The accuracy changes are marginal, with some configurations even slightly outperforming the original MLP, indicating that TransCoder integration does not degrade model capabilities. In addition to performance metrics, we also analyze the error patterns across different model variants. As shown in Appendix Table 14, the distribution of top error clusters remains largely unchanged

Table 5: **Comparison of Overall Accuracy Between CodeLlama and its Variants**

| Layer | MLP | LTC4 | LTC8 | LTC16 |
|---|---|---|---|---|
| layer 10 | 57.50% | 56.83% | 58.33% | 57.83% |
| layer 19 | 57.50% | 58.33% | 58.17% | 58.17% |
| layer 23 | 57.50% | 57.67% | 57.33% | 56.12% |
| **AVE** | 57.50% | 57.61% | 57.94% | 57.37% |

after replacing the MLP with TransCoder modules. This indicates that the core behavioral characteristics of the model are preserved. These results demonstrate the functional compatibility of the TransCoder module with standard MLP layers. It not only preserves predictive performance but also retains error-specific activation patterns, making it a reliable substitute for evaluating and manipulating the model's internal knowledge.

## 5 CONCLUSION

Existing locate-and-then methods are built upon the ROME approach. Although these methods improve performance, they still lack sufficient interpretability, making it difficult to understand how knowledge is injected into the MLP layers. Building on this, we propose TCPE, which combines ROME with TransCoder, revealing a clear correspondence between the edited neurons and the injected knowledge, thereby laying the groundwork for interpreting ROME-based methods such as MEMIT and PMET. On one hand, TCPE reveals a clear correspondence between the edited neurons and the injected knowledge. Through intuitive visualization, TCPE enables developers to transparently track the location of knowledge injection. On the other hand, TCPE's performance gains on TransCoder suggest that ROME-based methods' limited specificity stems from the polysemanticity of MLP neurons, as these edits can induce unintended interference beyond the target scope. This suggests a potential direction for future work: identifying and isolating sub-representations within polysemantic neurons that correspond to specific facts may be crucial for enhancing editing specificity.

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

APPENDIX

**Appendix Contents**

## A    RELATED WORK

**Code Large Language Models.** Recent advancements in Code LLMs have yielded substantial progress along four key dimensions: data quality, model architecture, training methodology, post-training refinement, and retrieval-augmented generation. On the **data-centric** side, efforts such as StarCoder (Li et al., 2023) introduced "The Stack v1.2", a high-quality, deduplicated, and permissively licensed code dataset, while Magicoder (Wei et al., 2024) proposed OSS-Instruct to synthesize diverse programming tasks from open-source repositories. Phi-1 (Gunasekar et al., 2023) demonstrated that compact yet high-quality synthetic corpora, such as "textbook-style" data, can yield strong model performance. Furthermore, MistralHermes-Code enhanced the capabilities of Mistral-7B (Jiang et al., 2023) through comprehensive multilingual fine-tuning on over 200,000 diverse code examples. With respect to **architectural and training** innovations, TransCoder (Rozière et al., 2020) integrated denoising auto-encoding and back-translation, marking the first successful application of unsupervised program translation in the code domain. Code Llama (Rozière et al., 2024) incorporated infilling-aware tokenizers and extends context length through RoPE scaling (Su et al., 2024), while LongCoder(Guo et al., 2023) addressed long-range dependencies via a sliding window attention mechanism (Clement et al., 2021).

In terms of **post-training** optimization, approaches like WizardCoder (Luo et al., 2024) employed Evol-Instruct for iterative instruction tuning, and OctoPack (Muennighoff et al., 2024) generated realistic development tasks based on GitHub commit histories. Reinforcement learning has also proven effective, CodeRL (Le et al., 2022) utilized unit tests and critic scores as reward signals, whereas RLTF fine-tuned models based on test pass rates (Liu et al., 2023). Gemma (Mesnard et al., 2024) combined Supervised Fine-Tuning (SFT) with Reinforcement Learning from Human Feedback (RLHF), enhancing its capability in natural language understanding for code-related tasks. Finally, **retrieval-augmented generation** techniques have emerged as a powerful paradigm. In this context, RepoCoder (Zhang et al., 2023) and CodeT5+ (Wang et al., 2023) combine retrieval mechanisms with pre-trained code language models to enhance diverse programming tasks.

However, Code LLMs are inherently static, and updating their internal knowledge to correct erroneous behavior remains a fundamental challenge. Conventional methods such as full retraining or fine-tuning are computationally intensive and prone to catastrophic forgetting (Luo et al., 2025; Li et al., 2024a; Zhu et al., 2024; GLM et al., 2024). Alternatives like prompt engineering or memory-based augmentation offer only superficial fixes, as they do not alter the model's internal representations (Wang et al., 2025b; Wang & Zhu, 2024; Zhang et al., 2025).

**Knowledge Editing Technology in NLP.** Knowledge editing aims to modify specific factual associations in pre-trained language models without retraining from scratch or degrading unrelated knowledge. Early works, such as MEND (Mitchell et al., 2022a), introduced a collection of small auxiliary editing networks for fast localized editing. Similarly, SERAC (Mitchell et al., 2022b) formulated editing as constrained generation and stored edits in an external memory, though it suffered from generalization issues as the memory size grew. Later, ROME (Meng et al., 2022) proposed directly intervening in the Transformer's feed-forward network (FFN) weights by computing rank-one updates, achieving strong editing performance with minimal side effects. Building on this, MEMIT (Meng et al., 2023) extended the idea to batched multi-edit settings by applying a series of localized updates to multiple layers, scaling the editing process while maintaining precision. Complementary to these editing-based approaches, LoRA (Hu et al., 2022) proposed a low-rank adaptation method for fine-tuning LLMs using low-rank matrices. While LoRA is not specifically designed for knowledge editing, it serves as a parameter-efficient fine-tuning baseline and has inspired hybrid approaches that blend fine-tuning and editing (Guo et al., 2025).

Recent work emphasized fine-grained localization and interpretability. PMET (Li et al., 2024b) built on this idea by analyzing the information flow within transformer layers, distinguishing between contributions from Multi-Head Self-Attention (MHSA), FFN, and residual paths (Miller et al., 2024). PMET found that MHSA primarily encoded general-purpose extraction behaviors and thus did not require modification. As a result, it restricted updates to FFN weights and used their corresponding hidden states for targeted editing. FiNE (Pan et al., 2025) proposed a neuron-level editing framework, locating and updating only a small number of FFN neurons, which improved the locality and precision of edits. Overall, recent trends in knowledge editing move from coarse-grained layer-wise updates to fine-grained neuron-level interventions.

However, most existing editing methods focus on manipulating FFN weights, where individual neurons often entangle multiple concepts or functions. This polysemantic nature (Scherlis et al., 2022) makes it difficult to interpret the effect of edits and to control them with high specificity (Elhage et al., 2021). As a result, even fine-grained interventions at the neuron level may inadvertently affect unrelated knowledge, limiting both interpretability and specificity. Such limitations highlight the importance of structured and interpretable representations as a foundation for precise and controllable model editing.

# B  LIMITATIONS AND BROADER IMPACT

**Limitations.** First, TCPE requires architectural modification of the LLM by replacing an MLP layer with a TransCoder. This may not be feasible in closed-source or production models and may introduce subtle behavioral changes. Second, the purpose of TCPE is to serve as a tool for understanding the underlying principles of knowledge editing: it focuses on interpreting the relationship between limited injected knowledge and its corresponding edits, and is not intended as a practical method for large-scale knowledge updating. Beyond these conceptual limitations, training TransCoders introduces additional memory and compute overhead, and the required sparse autoencoder training

does not trivially scale. This limits its direct applicability to large production LLMs. In addition, integrating TransCoder modules increases optimization complexity compared to standard MLP layers; while sparse activation patterns enhance interpretability, they can occasionally result in stochastic convergence behavior during gradient-based updates. With respect to empirical evaluation, although TCPE demonstrates consistent improvements in low-resource Java-to-D translation, future work should extend evaluations to other pairs of programming languages and a broader set of code scenarios in order to assess the generalizability and robustness of our results. Such studies would provide a deeper understanding of the method's applicability in various scenarios. Finally, TCPE's reliance on unit-test–based functional equivalence introduces dependencies on the quality of the test suites, and its behavior under repeated or large-scale edits remains unexamined. Future work should evaluate multi-edit accumulation, and sensitivity to the used set of unit tests.

**Broader Impact.** From a broader perspective, this work aims to advance safe and transparent knowledge editing for code-oriented large language models. The proposed neuron-level intervention mechanism facilitates the correction of model behaviors and the integration of new programming knowledge without exhaustive retraining, contributing to both open-source research and to practical deployment scenarios. Moreover, by minimizing interference with unrelated knowledge, TCPE provides a more interpretable and reliable model editing approach. To ensure ethical and responsible deployment, future work should establish auditing protocols, verification pipelines, and safeguard mechanisms to prevent unintended consequences of fine-grained model interventions.

## C   LLM USAGE STATEMENT

A large language model (LLM) was used solely to aid in polishing the text, improving phrasing, clarity, and readability. All scientific content, including ideas, experimental design, analysis, and conclusions, was developed entirely by the authors.

## D   BENCHMARKS AND BASELINES FOR KNOWLEDGE EDITING EVALUATION

### D.1   DATASETS

In this section, we provide detailed descriptions of the benchmarks used in our experiments: KECode (including the G4GD dataset), HumanEval, CounterFact, and zsRE.

- **G4GD** dataset is designed to evaluate knowledge editing in the context of low-resource code translation, with functional equivalence as the primary evaluation criterion. Many existing code benchmarks, such as MBPP (Austin et al., 2021), are less suitable for knowledge editing due to their small sample sizes and diverse task types. For example, tasks like "*Write a function to zip the two given tuples.*" are clearly irrelevant to knowledge editing. In contrast, we focus on code translation tasks because they are both a common benchmark in the code domain and provide a clear input–output mapping, which enables more direct observation of how knowledge edits affect model behavior. In this paper, the Java $\rightarrow$ D translation task was selected as an initial test case since D is a low-resource language, making the impact of knowledge edits on model performance more pronounced. This setup enables us to clearly observe how knowledge editing impacts both the targeted error clusters and the overall model accuracy.

- **HumanEval** (Chen et al., 2021) is a benchmark for code generation introduced by OpenAI. It consists of 164 hand-written programming problems, each paired with a natural language prompt and a hidden unit test. The dataset is specifically designed to measure functional correctness rather than superficial similarity, as solutions are automatically evaluated by executing the generated code against the ground-truth tests. HumanEval has become a standard benchmark for assessing the ability of LLMs to synthesize correct and executable programs from natural language descriptions. In our study, HumanEval is used to assess the overall impact of knowledge edits introduced in G4GD on model performance.

- **CounterFact** (Meng et al., 2022) is a large-scale dataset created to test factual knowledge editing in language models. Each instance in CounterFact contains a subject–relation–object triple, along with alternative target facts, paraphrased prompts, and counterfactual contexts.

The dataset is particularly useful for assessing whether a model can not only incorporate newly injected knowledge but also remain consistent when queried in diverse forms. Moreover, CounterFact provides auxiliary contexts that test whether the model can suppress its original memorized facts in favor of the new edits, making it a challenging and comprehensive benchmark for evaluating editing specificity and generalization.

- **zsRE** (Levy et al., 2017) is a question-answering dataset commonly used to evaluate the zero-shot generalization abilities of models. Each sample includes a question, an original answer, and a revised target answer, making it suitable for evaluating whether editing methods can effectively modify factual associations in LLMs. Compared with CounterFact, zsRE focuses more on the ability of a model to adapt to factual corrections in a question-answering setting, where robustness to paraphrasing and generalization across different query formulations are key evaluation aspects.

## D.2 EVALUATION METRICS FOR KNOWLEDGE EDITING IN NLP BENCHMARKS

For CounterFact and zsRE, we follow the experimental protocols of Pan et al. (2025), focusing on three key metrics: *Efficacy*, *Specificity*, and *Generalization*. Let $s_j$ denote the subject of a factual triple, $r_j$ the corresponding relation, and $o_j^*$ the target object. Furthermore, $p(s_j, r_j)$ represents the model's prompt constructed from $(s_j, r_j)$, $\mathcal{G}$ denotes the original model, and $\mathcal{G}'$ the post-edited model. Each metric is formally defined as follows.

- **Efficacy** quantifies whether the post-edited model produces the expected output and is formally defined as:

$$\text{Efficacy} = \mathbb{E}_{(s_j, r_j, o_j^*) \sim D_{\text{eff}}} \, \mathbb{I}\{\arg\max_y P_{\mathcal{G}'}[y \mid p(s_j, r_j)] = o_j^*\}.$$

- **Generalization** assesses the model's ability to propagate the edited knowledge to related real-world contexts:

$$\text{Generalization} = \mathbb{E}_{(s_j, r_j, o_j^*) \sim D_{\text{gen}}} \, \mathbb{I}\{\arg\max_y P_{\mathcal{G}'}[y \mid p(s_j, r_j)] = o_j^*\}.$$

- **Specificity** examines whether the edit affects only the targeted knowledge without influencing unrelated information. It can be quantified as:

$$\text{Specificity} = \mathbb{E}_{(s_j, r_j) \sim D_{\text{spe}}} \, \mathbb{I}\{\arg\max_y P_{\mathcal{G}'}[y \mid p(s_j, r_j)] = \arg\max_y P_{\mathcal{G}}[y \mid p(s_j, r_j)]\}.$$

## D.3 BASELINES

We compare our method against a set of representative knowledge-editing baselines:

- **ROME** (Meng et al., 2022) directly intervenes in the Transformer's feed-forward network weights via rank-one updates. By targeting specific neurons, it enables precise modification of factual knowledge with minimal side effects, particularly effective in single-edit scenarios. However, its specificity can be limited when multiple edits are required due to the polysemantic nature of neurons.

- **MEMIT** (Meng et al., 2023) extends ROME to multi-edit scenarios by applying localized updates across multiple layers. This enables the model to edit several factual associations in a single pass while maintaining high editing effectiveness, making it suitable for large-scale knowledge updates.

- **PMET** (Li et al., 2024b) improves precision in model editing by separating FFN-specific hidden states from those of MHSA, using only the FFN-relevant component to update FFN weights.

- **Fine-Tuning** (**FT**) (Zhu et al., 2020b) is a standard approach where part or all of the model weights are updated on new data. While FT can successfully inject new knowledge, it is parameter-intensive, computationally costly, and prone to unintentional degradation of unrelated knowledge, making it less targeted than specialized editing methods.

- **AGRACE** (Li et al., 2025) is a model editing method designed for code LLMs, which leverages an external memory and a contrastive learning mechanism to correct erroneous knowledge across multiple editing instances. However, its evaluation relies on text-based validation metrics, which do not align with the conventional evaluation principle for code LLMs, namely the functional equivalence principle.

- **Few-shot** (Parnami & Lee, 2022) methods rely on in-context learning, providing the model with a small set of demonstration examples at inference time to modify its output behavior. While these methods are lightweight and flexible, they do not permanently modify model weights and thus offer limited long-term effectiveness and precision.

- **FiNE** (Pan et al., 2025) targets neuron-level interventions across multiple MLP layers with high granularity, focusing on specific subspaces of active neurons, thereby achieving precise edits while minimizing cross-interference.

- **WISE** (Wang et al., 2025a) introduces a dual-memory framework, where the main memory preserves pretrained knowledge while a side memory stores edits. A router dynamically selects between the two, and a knowledge-sharding mechanism is employed to avoid conflicts during continual editing.

- **LoRA** (Hu et al., 2022) introduces trainable low-rank matrices to adapt pre-trained models efficiently. By freezing the original weights and updating only the lightweight adapters, LoRA achieves substantial reductions in computational and memory costs while preserving model performance. Although not originally designed for knowledge editing, it is widely adopted as a parameter-efficient fine-tuning baseline.

# E    TCPE PERFORMANCE RESULTS

## E.1    IMPACT OF KNOWLEDGE EDITING METHODS ON OVERALL MODEL PERFORMANCE

Following the injection of Java-D correction knowledge (Section 4.3), ROME-based methods modify the internal parameters of code LLMs. Table 6 shows how these edits affect the models' overall performance via HumanEval. The RE reflects the global impact of the edit on the model's overall accuracy, while $AC_{post}$ and $AC_{pre}$ denote the model's post- and pre-edit functional accuracy, respectively. In Table 6, TCPE maintains the original functional accuracy ($AC_{post} = AC_{pre}$), indicating minimal disruption to the model's overall code generation capabilities. In contrast, other ROME-based methods, such as ROME, MEMIT, and PMET, update thousands of neurons (11,008–55,040) and exhibit lower

Table 6: **Effects of Knowledge Editing on Code LLMs via HumanEval.**

| Method | RE↑ | $AC_{post}$ | $AC_{pre}$ | Neurons |
|---|---|---|---|---|
| ROME | 93.61 | 26.83 | 28.66 | 11,008 |
| MEMIT | 76.45 | 21.91 | 28.66 | 55,040 |
| PMET | 53.18 | 15.24 | 28.66 | 55,040 |
| TCPE (LTC4) | 100.00 | 29.27 | 29.27 | 69 |

max_new_tokens=2000, pass@1.

reliability with noticeable reductions in post-edit accuracy. These results underscore the advantages of TCPE's neuron-level approach. By performing highly selective updates on specific neurons, TCPE enables precise knowledge injection while minimizing unintended side effects on the model's broader behavior.

## E.2    INFORMATION-CARRYING CAPACITY OF ACTIVE NEURONS IN SINGLE-ERROR EDITING SCENARIO

In the single-error editing scenario, we further investigate the information-carrying role of neurons in knowledge editing to support precise modifications, employing TCPE to conduct conventional experiments on high-activation neurons alongside ablation studies targeting low-activation neurons on the G4GD dataset.

**Role of Active TransCoder Neurons in Single-Error Editing Scenario.** Based on the results in Table 7, we examine the performance of single error types $C_0$ and $C_8$ under different activation thresholds ($acv$). For both $C_0$ and $C_8$, updating only a small number of highly activated neurons (i.e., $acv > 0.1$) results in noticeable improvements in generalization and reliability. As the threshold is

lowered to $acv > 0.05$ to include more neurons, the improvement in generalization and reliability becomes limited. Moreover, when more low-activation neurons (i.e., $acv > 0$) are introduced, both generalization, specificity, and reliability experience varying degrees of decline. This emphasizes the importance of focusing updates on neurons with high activation to achieve effective knowledge injection. Consequently, precise targeting of these highly activated neurons is critical for achieving effective and reliable knowledge injection.

Table 7: **Role of Active TransCoder Neurons in Single-Error Editing Scenarios: Error Clusters $C_0$ and $C_8$.** In LTC4, only neurons with activation values ($acv$) exceeding the specified thresholds $\tau \in \{0, 0.01, 0.05, 0.08, 0.1\}$ are updated. The term "Updated Neurons" refers to the count of neurons that surpass each threshold.

| Method | Efficacy | | Specificity | | Reliability | | | Updated Neurons |
|---|---|---|---|---|---|---|---|---|
| | GN↑ | CD↓ | LoC↑ | DT↓ | RE↑ | $AC_{post}$ | $AC_{pre}$ | |
| LTC4 ($C_0$) | | | | | | | | |
| $acv > 0.1$ | 38.46 | 51.28 | 83.24 | 6.29 | 103.71 | 60.50 | 58.33 | 29 |
| $acv > 0.08$ | 43.59 | 41.03 | 82.35 | 6.86 | 103.71 | 60.50 | 58.33 | 32 |
| $acv > 0.05$ | 43.59 | 41.03 | 82.17 | 6.86 | 103.71 | 60.50 | 58.33 | 41 |
| $acv > 0.01$ | 41.03 | 46.15 | 82.71 | 6.86 | 103.43 | 60.33 | 58.33 | 54 |
| $acv > 0$ | 41.03 | 46.15 | 82.71 | 6.86 | 103.43 | 60.33 | 58.33 | 57 |
| LTC4 ($C_8$) | | | | | | | | |
| $acv > 0.1$ | 28.85 | 28.85 | 97.45 | 0.86 | 104.00 | 60.67 | 58.33 | 14 |
| $acv > 0.08$ | 34.62 | 15.38 | 95.62 | 1.43 | 106.00 | 61.83 | 58.33 | 17 |
| $acv > 0.05$ | 34.62 | 13.46 | 95.62 | 1.43 | 106.00 | 61.83 | 58.33 | 21 |
| $acv > 0.01$ | 28.85 | 13.46 | 95.07 | 1.71 | 105.14 | 61.33 | 58.33 | 29 |
| $acv > 0$ | 28.85 | 11.54 | 95.07 | 1.71 | 105.14 | 61.33 | 58.33 | 30 |

**Ablation Study of Active TransCoder Neurons in Single-Error Editing Scenario.** In this section, we conduct an ablation study to examine the role of active TransCoder neurons in the context of single-error editing. In Table 8, we observe that as highly activated neurons are excluded from the update set, the model's generalization ability completely collapses (GN = 0 across all thresholds). Even when a large number of neurons are updated (e.g., up to 16,370 in $C_8$), the model fails to exhibit any effective generalization. Although specificity and reliability metrics remain high, this is due to the lack of real intervention. These stable values suggest that the model has not made meaningful modifications and has failed to successfully perform the intended edits. Further analysis reveals that even when the number of updated low-activation and inactive neurons increases (e.g., from 16,327 to 16,355 in $C_0$, or from 16,354 to 16,370 in $C_8$), no improvement in generalization occurs. This suggests these neurons play a negligible role in the learning process. This highlights that error-correction knowledge is primarily carried by highly active neurons, while a large number of low-activation and inactive neurons carry little to no knowledge.

Table 8: **Ablation Study on the Role of Active TransCoder Neurons in Single-Error Editing Scenarios: Error Clusters $C_0$ and $C_8$.** In LTC4, only neurons with activation values ($acv$) below specified thresholds $\tau \in \{0, 0.01, 0.05, 0.08, 0.1\}$ are updated. Here, "Updated Neurons" represents the number of neurons that meet each threshold.

| Method | Efficacy | | Specificity | | Reliability | | | Updated Neurons |
|---|---|---|---|---|---|---|---|---|
| | GN↑ | CD↓ | LoC↑ | DT↓ | RE↑ | $AC_{post}$ | $AC_{pre}$ | |
| LTC4 ($C_0$) | | | | | | | | |
| $acv \le 0.1$ | 0.00 | 100.00 | 99.64 | 0.00 | 100.00 | 58.33 | 58.33 | 16,355 |
| $acv \le 0.08$ | 0.00 | 96.30 | 99.48 | 0.29 | 100.00 | 58.50 | 58.33 | 16,352 |
| $acv \le 0.05$ | 0.00 | 100.00 | 98.95 | 0.29 | 100.86 | 58.33 | 58.33 | 16,343 |
| $acv \le 0.01$ | 0.00 | 100.00 | 99.65 | 0.29 | 100.00 | 58.33 | 58.33 | 16,330 |
| $acv = 0$ | 0.00 | 100.00 | 98.54 | 0.86 | 99.71 | 58.50 | 58.33 | 16,327 |
| LTC4 ($C_8$) | | | | | | | | |
| $acv \le 0.1$ | 0.00 | 100.00 | 99.64 | 0.29 | 99.71 | 58.83 | 58.33 | 16,370 |
| $acv \le 0.08$ | 0.00 | 100.00 | 98.54 | 0.86 | 99.71 | 58.17 | 58.33 | 16,367 |
| $acv \le 0.05$ | 0.00 | 100.00 | 99.64 | 0.00 | 100.00 | 58.17 | 58.33 | 16,363 |
| $acv \le 0.01$ | 0.00 | 96.30 | 99.48 | 0.29 | 100.00 | 58.17 | 58.33 | 16,355 |
| $acv = 0$ | 0.00 | 100.00 | 99.64 | 0.29 | 100.00 | 58.17 | 58.33 | 16,354 |

### E.3 UNDERSTANDING LOW SPECIFICITY IN ROME-BASED KNOWLEDGE EDITING

Most existing locate-and-then editing methods are built upon ROME, yet they are hindered by ROME's limited specificity (Li et al., 2024b). To systematically investigate the underlying causes of this low specificity in ROME-based approaches, we conduct a comparative study of TCPE and other ROME-based methods, including ROME, MEMIT, and PMET, on the NLP benchmarks ZsRE and CounterFact. Since generalization and specificity are inherently trade-offs, in this study we primarily focus on the behavior of specificity under the premise of successful knowledge injection, rather than on generalization. As shown in Table 9, TCPE achieves significantly enhanced editing specificity, updating only approximately 0.2% of all neurons. Moreover, PMET demonstrates superior performance compared to ROME and MEMIT because, during knowledge injection, it discards irrelevant or redundant information, thereby reducing interference and improving editing precision. This indicates that updating MLP neurons may cause different concepts or knowledge fragments encoded within the same polysemantic neurons to interfere with each other, resulting in unintended side effects. In contrast, TCPE selectively identifies neurons most relevant to the target knowledge and modifies only these neurons, thereby minimizing interference and achieving higher specificity. In summary, the low specificity of ROME-based methods primarily arises from the polysemantic nature of neurons, which encode excessive information unrelated to the target knowledge injection. This suggests a potential direction for future improvement: identifying and isolating sub-representations within polysemantic neurons that correspond to specific facts may be key to further enhancing editing specificity.

Table 9: **Quantitative Evaluation of TCPE and ROME-based Methods on NLP Benchmarks.**

| Dataset | Metric | TCPE | PMET | MEMIT | ROME |
|---|---|---|---|---|---|
| zsRE | Eff. ↑ | $97.8_{\pm0.4}$ | $96.9_{\pm0.2}$ | $97.2_{\pm0.2}$ | $97.7_{\pm0.2}$ |
| | Gen. ↑ | $50.1_{\pm1.9}$ | $57.3_{\pm0.9}$ | $55.9_{\pm0.9}$ | $56.5_{\pm0.8}$ |
| | Spe. ↑ | $70.9_{\pm2.0}$ | $64.6_{\pm1.0}$ | $46.1_{\pm0.9}$ | $49.7_{\pm0.9}$ |
| | Neurons | 30~50 | 55,040 | 55,040 | 11,008 |
| CounterFact | Eff. ↑ | $99.5_{\pm0.1}$ | $99.1_{\pm0.1}$ | $99.2_{\pm0.1}$ | $99.5_{\pm0.1}$ |
| | Gen. ↑ | $49.1_{\pm1.2}$ | $51.8_{\pm1.1}$ | $63.1_{\pm1.1}$ | $59.7_{\pm1.0}$ |
| | Spe. ↑ | $62.5_{\pm1.1}$ | $58.7_{\pm0.7}$ | $43.3_{\pm0.7}$ | $46.9_{\pm0.6}$ |
| | Neurons | 30~50 | 55,040 | 55,040 | 11,008 |

TCPE: It leverages MTC4 with $acv > 1 * 10^{-4}$.

# F    TCPE INTERPRETABILITY RESULTS

## F.1    INTERPRETABLE, POTENTIALLY INTERPRETABLE, UNINTERPRETABLE, CONTEXT-FREE EXAMPLES

In this section, we provide some examples to support the analysis in Section 4.2.

▼ **Between 2.51 and 3.13: 0.0022%**
`max(a,b)>=2&&` Example 22721, token 29
`if(y[i])>=0&&(-` Example 20762, token 118
`if(name[0]==c)` Example 6588, token 11
`if(ss[i]ind)return` Example 18612, token 11
`ex_res[i]==0)` Example 9856, token 20
`if(node[0]==H-` Example 25127, token 11
`}]}]return` Example 7659, token 12
`1])/2)*(K-` Example 18341, token 59
`RD,RD);ans` Example 4459, token 105
`key(r,c)inq)continue` Example 18111, token 36

▼ **Between 1.88 and 2.51: 0.0707%**
`aset.keys[1]+1==` Example 14732, token 16
`}]}]returnd` Example 12741, token 26
`}]}]returnResult` Example 5977, token 12
`if(nx[i]<0)` Example 5751, token 83
`9;}}//writeln` Example 3445, token 10
`foreach(;0` Example 1784, token 96
`+1]));writeln` Example 3131, token 4
`1'){if(i<k` Example 47, token 37
`}}foreach(` Example 2545, token 17
`}]}]return` Example 877, token 9

▼ **Between 2.44 and 3.59: 0.0000%**
`y=uniform!int;sz` Example 13827, token 31

▼ **Between 1.28 and 2.44: 0.0167%**
`utable inf=int.max;}` Example 12400, token 19
`se 1;writ` Example 16909, token 73
`p[]=byte.max;d` Example 19055, token 22
`INF=long.max/3;` Example 18544, token 37
`INF=long.max/3;` Example 12073, token 31
`long x=long.min;` Example 12277, token 46
`int[]x){auto` Example 6317, token 23
`long ans=long.max;//` Example 11945, token 118
`long m=long.max;foreach` Example 9119, token 103
`+7;auto n=next` Example 824, token 66

▼ **Between 0.12 and 1.28: 11.1386%**
`7;long mod=9` Example 20136, token 5
`strings;scan(s);` Example 9638, token 53
`==cap)reserve(max(cap*` Example 2877, token 65
`]+a[1]<a[2` Example 2419, token 104
`int n=cast(int)s.length` Example 7833, token 13
`}import std.stdio,std.conv` Example 3012, token 90
`import std.stdio,std.conv` Example 131, token 106

(a) High-activation examples of a feature labeled as **interpretable**: This feature was annotated as a local context feature that activates when describing the closing part of paired symbols. For instance, tokens such as "}", ")", or "]" tend to trigger this feature.

(b) High-activation examples of a feature labeled as **Possibly-Interpretable**: This feature was annotated as a potential local context feature related to the semantic concept of "max". While it shows some consistent activation patterns, it remains unclear whether it reliably represents this concept.

▼ **Between 2.52 and 4.37: 0.0020%**
`:Andrei Alexandrescu 2` Example 6361, token 4
`i<n;i++){` Example 5907, token 20
`[0];autoY=input[` Example 25014, token 78
`!int();auto heights=new` Example 8193, token 99
`0];auto y=1[` Example 1604, token 100
`scanf("%d",&a[i]);` Example 5907, token 30

▼ **Between 0.68 and 2.52: 0.8873%**
`{long a,b,c,` Example 11724, token 97
`{int x,y,s;` Example 11975, token 114
`{long a,b;` Example 4287, token 103
`{int a,b,c,` Example 2217, token 64
`[1];autoC=abc[` Example 3143, token 54
`.to!(int[]);int a` Example 143, token 98
`{double x,y,r;` Example 88, token 89
`Copyright:Andrei Alexandrescu` Example 777, token 64
`min(A,B,C,D,` Example 1381, token 9
`i=a;i<=t;i+=` Example 11, token 65

▼ **Between -1.16 and 0.68: 99.0799%**
`line.popFront();}]` Example 43, token 10
`void moda(ref long x,long y` Example 123, token 38
`mod=9982443` Example 448, token 22

▼ **Between 6.28 and 7.85: 0.0003%**
`i++){th`=read` Example 35, token 8
`;auto th=new long[` Example 3597, token 95
`UM 90 TH THORI` Example 9274, token 82
`*2;j<th.length;j` Example 6093, token 21
`int[]th s;` Example 12649, token 105
`read();long th=x` Example 8010, token 125

▼ **Between 4.71 and 6.28: 0.0004%**
`500*th)/5` Example 7684, token 20
`;case"THU":` Example 21809, token 114
`4,"THU":3` Example 13058, token 78
`S.length*K/` Example 17601, token 13
`"WED","THU","F` Example 22043, token 100
`case"THU":writeln` Example 11131, token 95
`UE","WED","THU","FRI` Example 10291, token 41
`"WED","THU","F` Example 1238, token 113
`90 TH THORIUM` Example 9274, token 84

▼ **Between 3.14 and 4.71: 0.0018%**
`69 TM THULIUM` Example 11260, token 122
`class EOFException:Throwable{this` Example 5857, token 31
`class EOFException:Throwable{this` Example 25002, token 17

(c) High-activation examples of a feature labeled as **uninterpretable**: This feature was deemed uninterpretable because the text fragments that trigger it show no apparent semantic or structural consistency, making its meaning difficult to infer.

(d) High-activation examples of a feature labeled as **context-free**: This feature was annotated as a single-token feature, specifically activating on the occurrence of "th" in the middle of a word. It appears to fire independently of the broader linguistic context.

Figure 3: **Examples of "feature-dashboards" used in the feature interpretation experiments.**

## F.2 DETAILED RESULTS OF INTERPRETABILITY EXPERIMENTS ON ACTIVE NEURONS

In this section, we present detailed results that expand upon Section 4.2. In Figure 4, using all Java and D samples in $C_8$, we show some results from the top-10 most active features by examining their top-activating examples. These features consistently respond to key tokens such as 'str', 'string', or '=.length', which are directly associated with the corresponding error. (Additional results are given in the supplementary material.)

**▼ Between 3.54 and 4.43: 0.1221%**
`num){int len=num.length(` Example 0, token 64
`{int n1=str1.length;` Example 49, token 121
`;intn=str.length(` Example 65, token 27
`str){int n=str.length(` Example 73, token 14
`();int cl=corner.length(` Example 48, token 116
`str){int n=str.length;` Example 51, token 81
`intn=str.length()` Example 65, token 28
`str){int n=str.length(` Example 32, token 121

**▼ Between 2.66 and 3.54: 0.3743%**
`0;int n=tree.length;` Example 34, token 120
`corner){int n=str.length(` Example 48, token 107
`s){int n=s.length(` Example 47, token 101
`.length;int cl=corner.length;` Example 12, token 118
`str){int n=` Example 21, token 127
`s){int n=s.length(` Example 33, token 83
`n){int len=str.length;` Example 19, token 107
`for(int k=res.length-` Example 93, token 10
`s){int n=s.length;` Example 3, token 61
`);int cl=corner.length()` Example 48, token 117

**▼ Between 1.77 and 2.66: 0.5615%**
`=0,n=s.length;` Example 41, token 10
`str){int n=str.length;` Example 88, token 48
`){int n=str.length()` Example 73, token 15
`s){int l=s.length;` Example 20, token 85
`n=str.length();` Example 65, token 29

(a) Feature Index: 1787.

**▼ Between 3.30 and 3.96: 0.0000%**
`0;int n=tree.length()` Example 90, token 89

**▼ Between 2.64 and 3.30: 0.0326%**
`0;int n=tree.length()` Example 90, token 89
`0, j=str.length()` Example 78, token 17
`){int l=num.length;int` Example 58, token 103
`){int n=bin.length;if` Example 66, token 77

**▼ Between 1.98 and 2.64: 0.1790%**
`){int n=str.length()` Example 52, token 37
`charAt(start++);return result;` Example 23, token 122
`length;int cl=corner.length;if` Example 12, token 119
`0;i<str.length()` Example 27, token 98
`[i]-'0')*1` Example 10, token 60
`){int n=bin.length()` Example 42, token 47
`){int length=num.length;if` Example 43, token 46
`){int n=str.length()` Example 32, token 122
`){int n=str.length()` Example 48, token 108
`){int length=num.length()` Example 7, token 25

**▼ Between 1.32 and 1.98: 0.3906%**
`n-cl,n).equals(corner` Example 45, token 31
`_rot)||str1.equals(ant` Example 66, token 52
`pre_sum[i])/(n` Example 87, token 73
`for(int k=res.length-` Example 93, token 10
`){int n=str.length;int` Example 12, token 112
`){int len=str.length;int` Example 13, token 15

(b) Feature Index: 13321.

**▼ Between 8.58 and 10.72: 0.1546%**
`..str.length){` Example 25, token 2
`=str[start++];` Example 48, token 2
`int h=str.length-` Example 29, token 4
`==str.charAt(` Example 17, token 2
`=clock_rot+str2.substring(` Example 66, token 18
`j++){if(str[i]!=` Example 3, token 7
`{int n=str.length;return` Example 52, token 5
`j++){if(str[i-` Example 4, token 46
`;K++)if(str[i]==` Example 15, token 45
`N){if(str[i]==` Example 84, token 93

**▼ Between 6.43 and 8.58: 0.3418%**
`_filled(string[] str1,string[]` Example 58, token 8
`K++)if(str.charAt(` Example 86, token 28
`{res=res+str[i-` Example 85, token 86
`2){if(str1.length!=` Example 9, token 12
`f_filled(string str){int i` Example 5, token 8
`_count;res=str.charAt(` Example 88, token 23
`){int n=str.length;int` Example 77, token 18
`f_filled(string str){int n` Example 54, token 112
`_gold(String str){int i` Example 78, token 6
`){int i=str.length()` Example 23, token 18

**▼ Between 4.29 and 6.43: 0.9277%**
`f_filled(string str){int n` Example 2, token 100
`){int n=str.length()` Example 75, token 112
`f_filled(string str){int i` Example 11, token 38

(c) Feature Index: 8370.

**▼ Between 6.88 and 8.60: 0.0488%**
`int f_filled(string s){int` Example 31, token 7
`arr=new String[sub_count` Example 18, token 4
`string str1,string str2){` Example 9, token 5
`;return result;}string f_filled(` Example 65, token 105
`l][j]);string res="";int` Example 85, token 29
`[n];}string f_filled(` Example 2, token 94

**▼ Between 5.16 and 6.88: 0.1139%**
`char f_filled(string str){int` Example 77, token 11
`char f_filled(string str){foreach` Example 62, token 117
`_gold(String str){int` Example 78, token 5
`l];}string f_filled(` Example 86, token 113
`}return true){string f_filled(` Example 78, token 65
`l];}string f_filled(` Example 69, token 29
`int f_filled(string str){int` Example 51, token 75

**▼ Between 3.44 and 5.16: 0.2930%**
`string str1,string` Example 9, token 1
`int f_filled(string str)` Example 36, token 125
`int f_filled(string s)` Example 82, token 39
`){String s=String.valueOf(` Example 71, token 21
`bool f_filled(string str,string corner` Example 12, token 102
`bool f_filled(string str){int` Example 40, token 118
`}sort(arr);string res="";for` Example 33, token 38
`char f_filled(string str){int` Example 38, token 68
`int f_filled(string tree,int k` Example 34, token 99
`bool f_filled(string str,int n` Example 19, token 98

(d) Feature Index: 5579.

Figure 4: **Top-Activating Examples for Active Neurons.**

### F.3 Detailed Results of Interpretability Experiments on Inactive Neurons

This section presents detailed results that complement the analysis Section 4.2. Using all Java samples and D samples in $C_8$ as input, we capture and analyze the patterns of 10 randomly selected inactive features. As shown in Figure 5, although these features exhibit interpretability due to the sparsity induced by the TransCoder, they primarily attend to irrelevant tokens such as 'if', 'N', ';', and 'ps', indicating limited relevance to the target error cluster $C_8$. (Additional results are given in the supplementary material. As some inactive neurons fail to respond to relevant examples, only 8 results are included.)

▼ **Between 3.17 and 3.97: 0.0326%**
i=n, j=n; while ( Example 85, token 39
1); i--; j--;} Example 62, token 30
]) i--; else j--;} Example 62, token 62
return false; i++; j--;} Example 78, token 56
▼ **Between 2.38 and 3.17: 0.2930%**
false;} i++; j--;} Example 12, token 88
]) return false; i++; j--;} Example 5, token 48
][[j])}}}return Example 86, token 94
j−1]}}} for ( Example 80, token 112
)−'0'; group+=(num Example 22, token 27
n];int i, j, k, Example 61, token 25
0) count++;} return count; Example 36, token 114
true;} else i++;} return true; Example 41, token 65
!=k) return true;}}} return Example 1, token 101
j−1]; else dp[ Example 17, token 43
▼ **Between 1.59 and 2.38: 1.7090%**
][[j])}}} return d Example 2, token 80
i+1];} if (open Example 49, token 32
, i+len);}} sort (arr Example 33, token 32
1]+1; else auxArr[ Example 21, token 44
, i+n);} Arrays. Example 51, token 52

(a) Feature Index: 3990.

▼ **Between 2.05 and 3.07: 0.3337%**
=1; if (str. charAt Example 46, token 12
')level−−; else if (level==k Example 79, token 39
; i−−){ if (str[i Example 8, token 53
=str.length; if (len>=n Example 19, token 112
(')level++; else if (tree[i Example 79, token 26
count++;} if (cur_count> Example 88, token 7
=='1' && !oneSeen) Example 38, token 24
0){ if (str[i] Example 69, token 71
length−1; while (i<j Example 5, token 26
▼ **Between 1.02 and 2.05: 1.4160%**
i=0; i<n; Example 28, token 17
, j=n; while (i> Example 85, token 42
<j){ if (str[i] Example 5, token 34
num.length; if (length==1 Example 43, token 51
L−1; if (L== Example 74, token 55
n; i++){ if (tree[i Example 75, token 47
i<j){ if (str.char Example 78, token 33
i; j++){ if (P[j Example 19, token 43
; K++) if (str.charAt Example 86, token 27
0; i−−) if (s[i Example 91, token 38
▼ **Between 0.00 and 1.02: 3.0599%**
Seen=true; if (getChar!= Example 38, token 35
){ if (P[0] Example 32, token 5

(b) Feature Index: 51.

▼ **Between 0.63 and 0.79: 0.0163%**
++){ strings=N. substring((0 Example 6, token 20
_gold (StringN){int len Example 81, token 112
▼ **Between 0.47 and 0.63: 0.0488%**
); String t=N. substring(i Example 36, token 57
){ String s=N. substring( Example 36, token 32
&& str. substring(n−cl, n Example 11, token 22
&& str. substring(n−cl, n Example 45, token 26
length; string t=N. substring(i Example 6, token 39
f_filled (stringn){stringres Example 86, token 119
▼ **Between 0.31 and 0.47: 0.0326%**
){ int len=N. length() Example 81, token 118
[0][N−1] Example 69, token 22
if (n1!=n2) return false Example 58, token 40
; if (k<N){ if ( Example 94, token 31
▼ **Between 0.16 and 0.31: 0.0407%**
+num;} return num;} static int Example 64, token 91
returnC[0][n− Example 0, token 39
{String s=N. substring((0 Example 36, token 33
=n. length−1; j>= Example 60, token 9
} else {num=num. substring( Example 64, token 47

(c) Feature Index: 8772.

▼ **Between 1.16 and 1.45: 0.0081%**
(str[i]. isUpperCase) return Example 25, token 13
▼ **Between 0.87 and 1.16: 0.0163%**
1; else c ps[i][ Example 94, token 91
str[k]) c ps[i][k Example 84, token 103
▼ **Between 0.58 and 0.87: 0.0407%**
(n1!=n2) return false; Example 58, token 41
1; else c ps[i][k Example 92, token 10
k]=cps[i][ Example 94, token 63
}}} return c ps[0] Example 69, token 16
▼ **Between 0.29 and 0.58: 0.0732%**
length()−1; k>= Example 62, token 84
[k]=c ps[i][ Example 94, token 100
'1'){int posFromRight=n Example 54, token 7
[i]=auxArr[i− Example 91, token 5
[][]c ps=new int[ Example 84, token 4
[i]=concat. substring(i Example 16, token 21
1; if (posFromRight% Example 54, token 20
(k)) c ps[i][ Example 94, token 53
▼ **Between 0.00 and 0.29: 0.0407%**
0&& i!=j && j!=k Example 27, token 57
}sum=abs(sum); return (sum Example 10, token 87

(d) Feature Index: 9831.

Figure 5: **Top-Activating Examples for Inactive Neurons.**

# G ACTIVE NEURON OVERLAP ANALYSIS

To evaluate the overlap of active neurons ($acv > 0$) across different error clusters, we select multiple clusters $C_i$, where $i \in \{0, 1, 2, 4, 6, 8, 9, 15\}$. For each cluster $C_i$, we construct five correction knowledge four-tuple $(r_{(j,i)}^{(1)}, s_{(j,i)}, r_{(j,i)}^{(2)}, o_{(j,i)}^*)$, where $j \in [1, 5]$, and compute the corresponding representation vectors $k_{(j,i)}^*$ (see Equation 6). Each $k_{(j,i)}^*$ is associated with a prefix set $\{a_{(j,i)}^n \mid n \in [1, 20]\}$, which includes ten prefixes of length 5 and ten of length 10. Based on this, we analyze the overlap of active neurons at layer 19 for the MLP of CodeLlama and the TransCoder modules of LTC4, LTC8, and LTC16.

## G.1 METRICS FOR NEURON ACTIVATION OVERLAP ACROSS ERROR CLUSTERS

In this section, we quantify the overlap of neuron activation patterns across different error clusters. We define two metrics: (1) **Absolute Overlap (AO)**: The ratio between the union of all activated neurons across clusters and the sum of neuron unions within each individual cluster. A higher value indicates greater independence between clusters. (2) **Relative Overlap (RO)**: The ratio between the total union of activated neurons and the average number of activated neurons per cluster, reflecting the degree to which active neurons are shared across clusters. Formally, these metrics are defined as:

$$\text{AO} = \frac{\left| \bigcup_{i=1}^{N} \bigcup_{j=1}^{k} S_{i,j} \right|}{\sum_{i=1}^{N} \left| \bigcup_{j=1}^{k} S_{i,j} \right|}, \quad \text{RO} = \frac{\left| \bigcup_{i=1}^{N} \bigcup_{j=1}^{k} S_{i,j} \right|}{\frac{1}{Nk} \sum_{i=1}^{N} \sum_{j=1}^{k} |S_{i,j}|}$$

Here, $S_{i,j}$ denotes the set of activated neuron indices for the $j$-th prompt in cluster $C_i$. In addition, we denote the average number of activated neurons across clusters as $A_1$, and within clusters as $A_2$. Their corresponding union sets are $U_1$ (cross-cluster) and $U_2$ (intra-cluster). The intersection of active neurons within a cluster is defined as $I_2$.

## G.2 ANALYSIS OF ACTIVE NEURON OVERLAP BETWEEN MLP AND TRANSCODER

In Table 11, we show the overlap of active neurons in the $k_{(j,i)}^*$ generated by multiple correction knowledge $(r_{(j,i)}^{(1)}, s_{(j,i)}, r_{(j,i)}^{(2)}, o_{(j,i)}^*)$ within the same cluster $C_i$. Obviously, the active neurons in the TransCoder module are more concentrated compared to those in the MLP for the same error type. As shown in Table 10, in the MLP layer, the RO value is 1.97 and the AO value is 0.16, indicating a concentrated distribution of active neurons with substantial overlap across error clusters. In contrast, the TransCoder neurons exhibit stronger independence across clusters, suggesting more distinct representations of different errors. The AO and RO values for the TransCoder module consistently remain high, with AO ranging from 0.41 to 0.47 and RO from 6.67 to 8.02. This consistency reflects a more structured and sparse organization of active TransCoder neurons, demonstrating the potential for fine-grained knowledge editing.

Table 10: **Comparison of Cross-Cluster Active Neuron Overlap between MLP and TransCoder Modules.** In the MLP layer, lower Absolute Overlap (AO) and Relative Overlap (RO) values indicate a more distributed set of active neurons with substantial overlap across error clusters. In contrast, the TransCoder layer consistently exhibits higher AO and RO values, reflecting more distinct and functionally independent active neurons across clusters.

| layer | $A_1$ | $U_1$ | RO | AO |
|-------|-------|-------|------|------|
| MLP   | 5514  | 10836 | 1.97 | 0.16 |
| LTC4  | 117   | 848   | 7.25 | 0.44 |
| LTC8  | 132   | 1059  | 8.02 | 0.47 |
| LTC16 | 448   | 2987  | 6.67 | 0.41 |

Table 11: **Intra-Cluster Active Neuron Counts and Overlaps for MLP and TransCoder Modules.** Obviously, the TransCoder modules (LTC4, LTC8, LTC16) exhibit significantly more localized and concentrated active neurons (see $A_2$ and $U_2$) compared to the MLP in CodeLlama.

|  | **Stats** | $C_0$ | $C_1$ | $C_2$ | $C_4$ | $C_6$ | $C_8$ | $C_9$ | $C_{15}$ | **AVE** |
|---|---|---|---|---|---|---|---|---|---|---|
| **CodeLlama** | $A_2$ | 6455 | 7502 | 9144 | 9279 | 9305 | 7548 | 8214 | 7413 | 8108 |
|  | $I_2$ | 4543 | 3538 | 1841 | 1822 | 1715 | 3449 | 2807 | 3646 | 2920 |
|  | $U_2$ | 5495 | 5520 | 5470 | 5521 | 5500 | 5532 | 5533 | 5538 | 5514 |
| **LTC4** | $A_2$ | 138 | 111 | 108 | 116 | 124 | 113 | 109 | 115 | 117 |
|  | $I_2$ | 95 | 51 | 20 | 20 | 25 | 51 | 34 | 64 | 45 |
|  | $U_2$ | 191 | 185 | 270 | 306 | 339 | 200 | 236 | 202 | 241 |
| **LTC8** | $A_2$ | 147 | 131 | 127 | 134 | 138 | 126 | 122 | 131 | 132 |
|  | $I_2$ | 91 | 67 | 25 | 29 | 29 | 61 | 43 | 63 | 51 |
|  | $U_2$ | 223 | 214 | 328 | 369 | 389 | 232 | 284 | 233 | 284 |
| **LTC16** | $A_2$ | 551 | 452 | 427 | 440 | 430 | 424 | 396 | 454 | 447 |
|  | $I_2$ | 386 | 236 | 73 | 92 | 76 | 198 | 123 | 263 | 181 |
|  | $U_2$ | 759 | 736 | 1068 | 1172 | 1208 | 746 | 885 | 754 | 916 |

# H   FINE-GRAINED CAUSAL INTERVENTION

Based on the causal intervention technique proposed in ROME (Meng et al., 2022), we introduce a *fine-grained causal intervention method* for token-by-token analysis. Specifically, we first pass a input $x$ into the model $G$ and collect all **clean activations** $\{h_i^{(l)} | i \in [1, T], l \in [1, L]\}$ across layers. Then, we sequentially corrupt a single token $i^*$ by adding noise at the embedding layer $h_i^{(0)} := h_i^{(0)} + \varepsilon$. This yields a set of **corrupted activations** $\{h_{i*}^{(l)} | i \in [1, T], l \in [1, L]\}$ throughout the network. Next, at each layer, we restore the clean hidden activation $h_i^{(l)}$ for the selected token, while leaving the rest of the activations corrupted. We measure the difference in model outputs between the restored and non-restored cases, producing a table of size $T \times L$, where each cell quantifies the causal effect of restoring the hidden state of a specific token at a specific layer.

Building on this procedure, we leverage the fine-grained causal intervention method to systematically assess the contribution of each token in the four-tuple $(r^{(1)}, s, r^{(2)}, o)$ across various programming language scenarios. As illustrated in Figure 6, our analysis reveals that the subject token $s$ exerts a substantial causal influence on the prediction of the object token $o$.

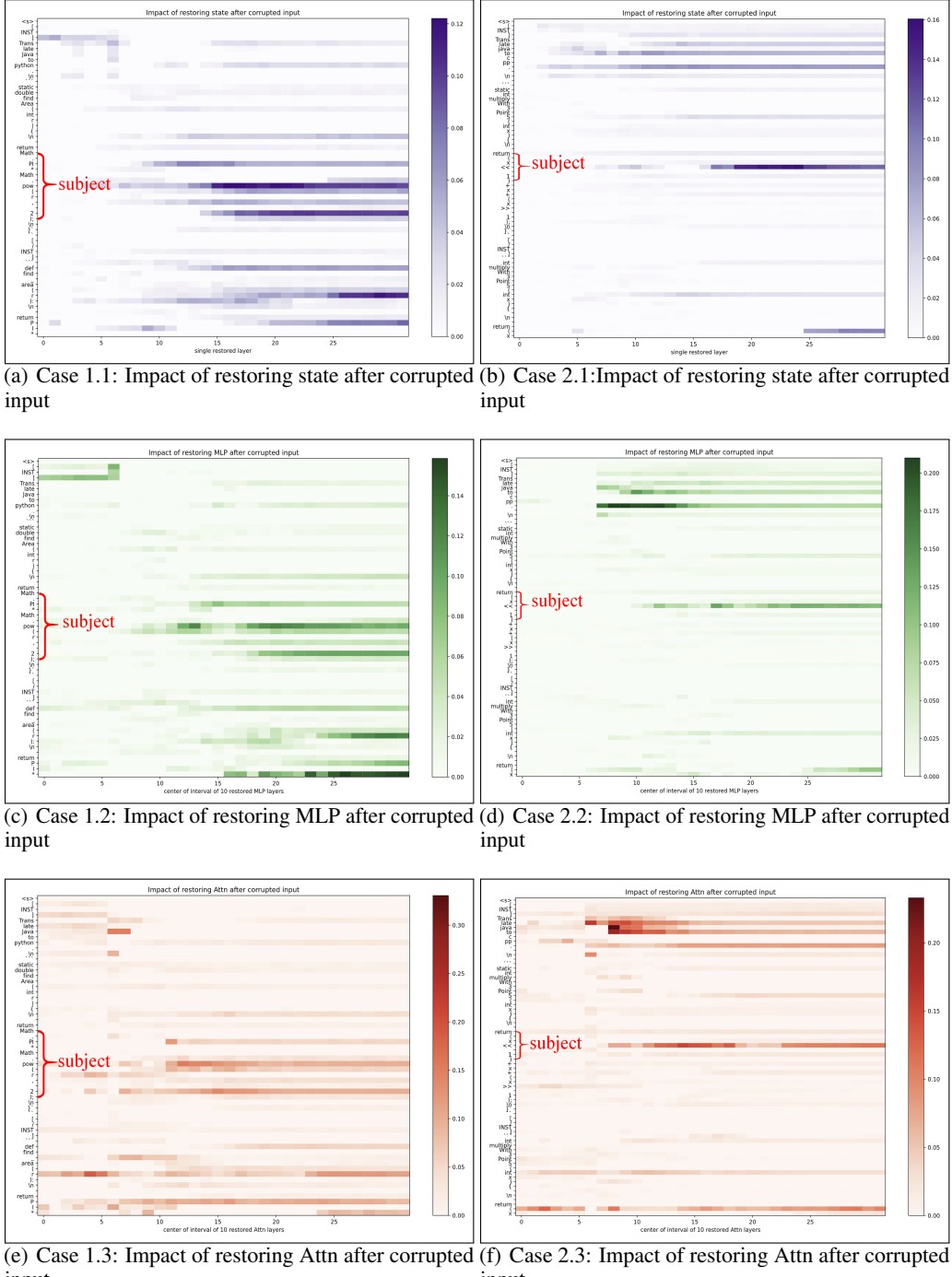

(a) Case 1.1: Impact of restoring state after corrupted input

(b) Case 2.1:Impact of restoring state after corrupted input

(c) Case 1.2: Impact of restoring MLP after corrupted input

(d) Case 2.2: Impact of restoring MLP after corrupted input

(e) Case 1.3: Impact of restoring Attn after corrupted input

(f) Case 2.3: Impact of restoring Attn after corrupted input

Figure 6: **Analyzing Each Token Behavior via Fine-grained Causal Intervention in CodeLlama.**

# I  CONSTRUCTING CORRECTION KNOWLEDGE FOUR-TUPLES

In this section, we introduce the process of constructing correction knowledge four-tuples $(r^{(1)}, s, r^{(2)}, o)$ for fixing specific error types in Java-to-D programming language translation. More specifically, the process begins by identifying translation failures in the Java-to-D task. For example, when translating Java functions to D, the translation fails the unit test. Leveraging the associated error messages, we locate the incorrect code snippet $o$ and construct the initial four-tuples $(r^{(1)}, s, r^{(2)}, o)$, where $o$ is the prediction generated by the model based on the prompt $p(r^{(1)}, s, r^{(2)})$. We then manually correct $o$ to the correct object $o^*$, forming the correction knowledge four-tuple $(r^{(1)}, s, r^{(2)}, o^*)$.

In Figure 7, we provide an example. In the Java-to-D translation, we consider a prompt $p(r^{(1)}, s, r^{(2)})$, where the source Java code snippet is $s =$ "`str.length`", and the model generates the D code snippet $o =$ "`str.length()`". Here, the generated $o$ is incorrect and does not conform to D language rules. We refine $o$ into the properly formatted D code snippet $o^* =$ "`cast(int)str.length()`", thereby constructing the correction knowledge four-tuple $(r^{(1)}, s, r^{(2)}, o^*)$.

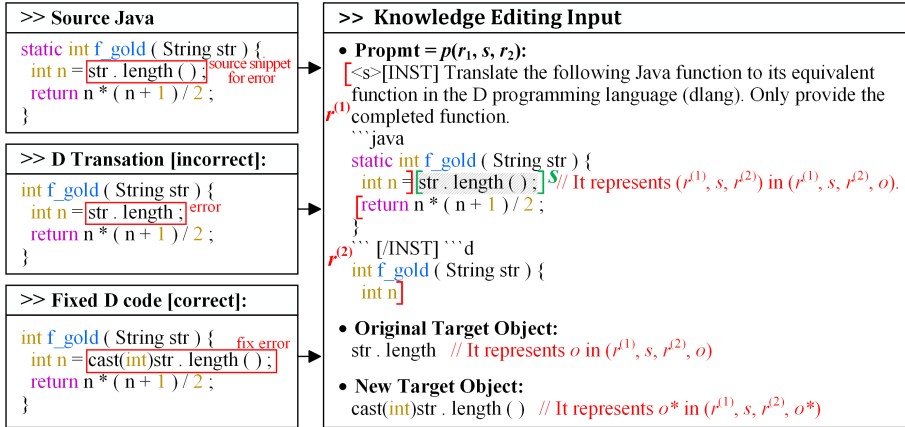

Figure 7: **Construction of Correction Knowledge Four-Tuple** $(r^{(1)}, s, r^{(2)}, o^*)$ **in Java-to-D Code Translation.**

# J  G4GD DATASET DETAILS

In this section, we provide a comprehensive overview of the G4GD dataset, including its structure, evaluation with standard benchmarks (MBPP and HumanEval), and representative examples. In addition, we present a detailed error analysis of CodeLlama in the Java-D translation task, covering a complete list of error categories, clustering statistics, intra-cluster examples, and comparisons across model variants.

## J.1  RATIONALE FOR LOW-RESOURCE TRANSLATION IN KNOWLEDGE EDITING EVALUATION

Popular existing datasets, such as HumanEval (Chen et al., 2021) and MBPP (Austin et al., 2021), have been widely used to evaluate the performance of code generation models. However, due to the small sample size and the diversity of task types, these datasets are not well suited for knowledge editing. For example, a task such as "*Write a python function to identify non-prime numbers.*" are clearly irrelevant to knowledge editing.

In this paper, we instead focus on code translation tasks, which are widely used benchmarks in the code domain and provide explicit input–output mappings, enabling a direct analysis of the effects of knowledge editing. As an initial case study, we adopt the Java → D translation task. Since D is a low-resource language, knowledge edits are more likely to produce observable changes, allowing us to precisely assess their influence on both specific error clusters and the overall model accuracy. We develop a specialized benchmark dataset, G4GD, designed to evaluate low-resource Java-to-D

language translation. The G4GD dataset comprises 600 Java functions, each paired with a suite of D unit tests. Every test suite includes 10 independent test cases designed to assess the functional correctness of the translated D code (see Table 16 for an example).

## J.2 EVALUATION OF G4GD DATASET WITH MBPP AND HUMANEVAL

To comprehensively evaluate the G4GD dataset, we compared it with existing standard benchmarks, HumanEval and MBPP. In terms of sample size, the HumanEval dataset contains only 164 examples (each accompanied by 7.7 automated test cases (Chen et al., 2021)), while the MBPP dataset includes 974 examples (each with 3 automated test cases (Austin et al., 2021)). In contrast, the G4GD dataset includes 600 examples, with a richer and more comprehensive set of unit test cases, offering a more robust and diverse resource for model training and evaluation. Furthermore, as shown in Figure 8, the code snippets in the G4GD dataset are generally longer, which presents a greater challenge in testing knowledge editing methods for handling complex functions. This design enables the dataset to more effectively assess the model's knowledge editing capabilities in more complex scenarios, particularly in the translation of long code snippets and complex functions, highlighting the model's robustness and adaptability.

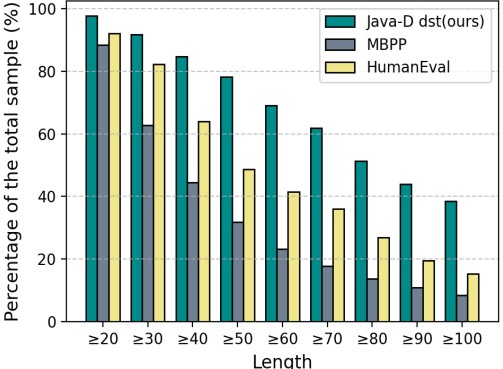

| Length | G4GD (ours) | MBPP | HumanEval |
|---|---|---|---|
| len $\geq$ 20 | 97.67% | 88.5 % | 92.07 % |
| len $\geq$ 30 | 91.83 % | 62.83 % | 82.32 % |
| len $\geq$ 40 | 84.67 % | 44.56 % | 64.02 % |
| len $\geq$ 50 | 78.17 % | 31.83 % | 48.78 % |
| len $\geq$ 60 | 69.17 % | 23.20 % | 41.46 % |
| len $\geq$ 70 | 61.83 % | 17.76 % | 35.98 % |
| len $\geq$ 80 | 51.33 % | 13.66 % | 26.83 % |
| len $\geq$ 90 | 44.00 % | 10.88 % | 19.51 % |
| len $\geq$ 100 | 38.50 % | 8.52 % | 15.24 % |

Figure 8: **Comparison of Input Sequence Length Distributions in G4GD, HumanEval, and MBPP.** We compare the input code length distributions of G4GD with widely used benchmarks HumanEval and MBPP. Obviously, G4GD contains significantly longer sequences, making it better suited for evaluating knowledge editing on more complex functions. Here, the $x$-axis denotes sequence length ranges, and the $y$-axis indicates the proportion of samples.

### J.3 Complete List of Error Messages by Cluster in CodeLlama

To characterize failure patterns, we first present a complete error message list, as summarized in Table 12, where each error cluster $C_i$ corresponds to a distinct type of compilation failure. This systematic categorization serves as the basis for subsequent error clustering, wherein all failed translation outputs are grouped according to their respective error types.

Table 12: Complete List of Error Categories for Codellama in Java-to-D Translation

| Cluster | Error Message |
|---------|---------------|
| $C_0$ | Error: instead of C-style syntax, use D-style 'int[][] mat' |
| $C_1$ | Error: 'std.math.algebraic.sqrt' called with argument types '(int)' matches both: |
| $C_2$ | Error: found '>' when expecting ';' following statement 'Set < (int)' |
| $C_4$ | Error: C style cast illegal, use 'cast(int)x' |
| $C_5$ | Error: '1 == 0' must be surrounded by parentheses when next to operator '&' |
| $C_6$ | Error: undefined identifier |
| $C_7$ | Error: none of the overloads of template 'std.algorithm.sorting.sort' |
| $C_8$ | Error: cannot implicitly convert expression 's.length' of type 'ulong' to 'int' |
| $C_9$ | Error: identifier expected following comma |
| $C_{10}$ | Error: semicolon expected following auto declaration, not '>' |
| $C_{11}$ | Error: can only '*' a pointer, not a 'int' |
| $C_{12}$ | Error: expression expected, not ')' |
| $C_{13}$ | Error: incompatible types for '(startIndex) + ("to")': 'int' and 'string' |
| $C_{14}$ | Error: no property 'substring' for 'first' of type 'string' |
| $C_{15}$ | Error: variable 'n' cannot be read at compile time |
| $C_{16}$ | Error: template instance 'HashSet!int' template 'HashSet' is not defined |
| $C_{17}$ | TestRuntimeError |
| $C_{18}$ | Error: invalid array operation |
| $C_{19}$ | Error: 'switch' statement without a 'default' |
| $C_{20}$ | Error: no identifier for declarator 'char' |
| $C_{21}$ | Error: missing closing ')' |
| $C_{22}$ | Error: slice 's' is not mutable |
| $C_{23}$ | Error: unterminated character constant |
| $C_{24}$ | Error: function is not callable |
| $C_{25}$ | Error: template argument expected following '!' |
| $C_{26}$ | Error: integer overflow |
| $C_{27}$ | Error: '10.0' is not of integral type, it is a 'double' |

### J.4 Error Cluster Statistics and Intra-Cluster Examples in CodeLlama

In this section, we analyze the distribution and characteristics of compilation failures produced by CodeLlama in Java-to-D translation. We cluster all failed translation outputs based on their corresponding error types. The resulting distribution, shown in Figure 9, reveals a highly concentrated pattern: the top four clusters ($C_0$, $C_8$, $C_4$, $C_6$) account for 58.51% of all compilation failures. Table 13 provides representative examples from two of the most prominent clusters: $C_0$ and $C_4$. Here, cluster $C_0$ contains violations of D-style array declaration conventions, and $C_4$ captures illegal uses of C-style casts.

Table 13: Examples of Error Clusters "$C_0$" and "$C_4$" in Java-to-D Translation for CodeLlama

| Cluster | Error Message | D Translation File |
|---------|---------------|--------------------|
| $C_0$ | Error: instead of C-style syntax, use D-style 'int[][] mat' | COUNT_SORTED_ROWS_MATRIX.d |
| | Error: instead of C-style syntax, use D-style 'int[n + 1] dp' | FRIENDS_PAIRING_PROBLEM.d |
| | Error: instead of C-style syntax, use D-style 'int[n + 1] dp' | LEONARDO_NUMBER_1.d |
| | Error: instead of C-style syntax, use D-style 'int[][] mat' | MAXIMUM_XOR_VALUE_MATRIX.d |
| | Error: instead of C-style syntax, use D-style 'int[][] LCStuff' | LONGEST_COMMON_SUBSTRING.d |
| | Error: instead of C-style syntax, use D-style 'int[n] ugly' | UGLY_NUMBERS.d |
| $C_4$ | Error: C style cast illegal, use 'cast(int)floor(digits)' | COUNT_DIGITS_FACTORIAL_SET_1.d |
| | Error: C style cast illegal, use 'cast(int)Math.sqrt(n)' | MINIMUM_PERIMETER_N_BLOCKS.d |
| | Error: C style cast illegal, use 'cast(int)Math.sqrt(ySquare)' | CIRCLE_LATTICE_POINTS.d |
| | Error: C style cast illegal, use 'cast(int)str1[i]' | SUM_TWO_LARGE_NUMBERS.d |
| | Error: C style cast illegal, use 'cast(int)sum' | EVEN_FIBONACCI_NUMBERS_SUM.d |
| | Error: C style cast illegal, use 'cast(int)str[i]' | LEXICOGRAPHICALLY_NEXT_STRING.d |

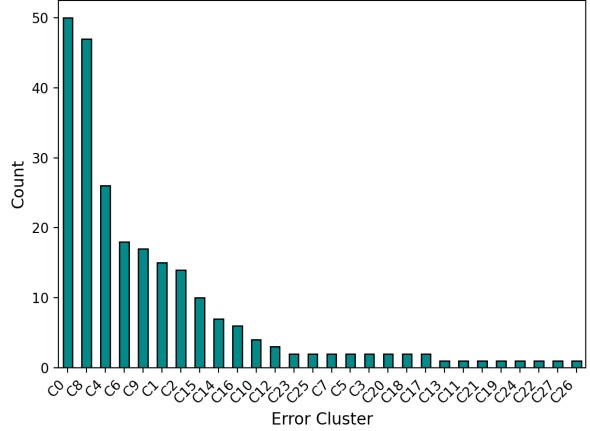

| Cluster | Count | Cluster | Count |
|---------|-------|---------|-------|
| $C_0$ | 50 | $C_5$ | 2 |
| $C_8$ | 47 | $C_3$ | 2 |
| $C_4$ | 26 | $C_{20}$ | 2 |
| $C_6$ | 18 | $C_{18}$ | 2 |
| $C_9$ | 17 | $C_{17}$ | 2 |
| $C_1$ | 15 | $C_{13}$ | 1 |
| $C_2$ | 14 | $C_{11}$ | 1 |
| $C_{15}$ | 10 | $C_{21}$ | 1 |
| $C_{14}$ | 7 | $C_{19}$ | 1 |
| $C_{16}$ | 6 | $C_{24}$ | 1 |
| $C_{10}$ | 4 | $C_{22}$ | 1 |
| $C_{12}$ | 3 | $C_{27}$ | 1 |
| $C_{23}$ | 2 | $C_{26}$ | 1 |
| $C_{25}$ | 2 | $C_{\text{succ}}$ | 345 |
| $C_7$ | 2 | $C_{\text{FailPass}}$ | 14 |

Figure 9: **Error Clusters of Codellama on Java-to-D Translation Tasks.** By applying error clustering to these compilation failure samples, we found that a significant portion of errors is concentrated in specific clusters. A notable concentration of errors is observed, with the top four clusters $C_0$, $C_8$, $C_4$, and $C_6$ together accounting for 58.50% of all compilation failures.

### J.5 COMPARISON OF TOP-8 ERROR CLUSTERS IN CODELLAMA, LTC4, AND LTC8

In this section, we compare the top-8 error clusters generated by CodeLlama and its two TransCoder variants, LTC4 and LTC8, which replace the 19th MLP layer with TransCoder modules of varying widths. As shown in Table 14, the distribution of the predominant error clusters remains largely consistent across all models. This stability suggests that the TransCoder effectively approximates the activation patterns of the original MLP layer, preserving the model's behavior despite architectural modifications.

Table 14: **Top-8 Error Clusters for Pure CodeLlama and CodeLlama Variants.** Here, each $C_i$ denotes a specific error type, with the frequency indicated in parentheses. LTC4 and LTC8 refer to CodeLlama variants where MLP layer 19 is replaced by a TransCoder module with $4,096 * 4$ and $4,096 * 8$, respectively.

| No | MLP | LTC4 | LTC8 |
|----|-----|------|------|
| 1 | $C_0$ (50) | $C_8$ (52) | $C_8$ (52) |
| 2 | $C_8$ (47) | $C_0$ (39) | $C_0$ (39) |
| 3 | $C_4$ (26) | $C_4$ (27) | $C_4$ (27) |
| 4 | $C_6$ (18) | $C_6$ (22) | $C_6$ (21) |
| 5 | $C_9$ (17) | $C_9$ (18) | $C_9$ (21) |
| 6 | $C_1$ (15) | $C_{10}$ (12) | $C_{10}$ (13) |
| 7 | $C_2$ (14) | $C_1$ (11) | $C_1$ (12) |
| 8 | $C_{15}$ (10) | $C_2$ (10) | $C_2$ (7) |
| 9 | $C_{\text{FailPass}}$ (14) | $C_{\text{FailPass}}$ (10) | $C_{\text{FailPass}}$ (10) |
| 10 | $C_{\text{succ}}$ (345) | $C_{\text{succ}}$ (350) | $C_{\text{succ}}$ (349) |

## J.6 EXAMPLES OF PROMPT AND OUTPUT

We provide a prompt that guides the model to translate code into an equivalent function in a target language. Table 15 illustrates a specific example where the model is prompted to translate a Java function into its equivalent implementation in the D programming language. The input prompt, enclosed in an " [INST]...[/INST] " block, specifies the task (i.e., translation from Java to D), followed by the source code. The model's output is the corresponding translation in the target language.

Table 15: **Example Prompt and Corresponding Output: Translating a Java Function to the D Language.**

| Type | Details |
|------|---------|
| Prompt & Output | ```
 [INST] Translate the following Java function to its equivalent
    function in the D programming language (dlang). Only provide the
    completed function.
```int solution (String s) {
    int result = 0;
    int n = s.length ();
    for (int i = 0;  i < n;  i++)
        for (int j = i;  j < n;  j++)
            if (s.charAt (i) == s.charAt (j))
                result++;
    return result;
}
``` [/INST] ```d
int solution(string s) {
    int result = 0;
    int n = cast(int) s.length;
    for (int i = 0; i < n; i++) {
        for (int j = i; j < n; j++) {
            if (s[i] == s[j])
                result++;
        }
    }
    return result;
}
``` 
``` |

## J.7 EXAMPLE FROM THE G4GD DATASET

To illustrate the structure of our dataset, Table 16 presents a representative example from the G4GD benchmark. Each entry in G4GD consists of a Java function paired with 10 independently constructed D unit tests. These tests are designed to evaluate the correctness of translated code across diverse inputs.

Table 16: **Example from the G4GD dataset:** The final G4GD dataset comprises 600 Java functions, each paired with 10 independent D unit tests.

| Type | Details |
|------|---------|
| Java function | ```
def f_gold ( n ) :
    if ( n == 0 or n == 1 ) :
        return n
    f1 , f2 , f3 = 0 , 1 , 1
    while ( f3 <= n ) :
        f1 = f2
        f2 = f3
        f3 = f1 + f2
    return f2
``` |
| D unit tests | ```
import std.stdio;
import std.math;
import std.conv;
import std.algorithm;

//TOFILL//

void main(){
    int [] results = [34, 55, 55, 55, 89, 34, 55, 89, 89, 55];
    int [] param0 = [54, 71, 64, 71, 96, 43, 70, 94, 95, 69];

    int n_success = 0;

    for (int i = 0; i < param0.length; i++) {
        if (results[i] == f_filled(param0[i])) {
                n_success += 1;
            }}

    writefln("#Results:%d,%d", n_success, param0.length);
}
``` |

## K  TRAINING DETAILS FOR KNOWLEDGE EDITING METHODS

### K.1  HYPERPARAMETER SETTINGS FOR CODE TASK

**ROME** The ROME configuration follows the original configuration proposed in the ROME[12], with a learning rate of 0.5, mom2_n_samples of 100,000, and a clamp_norm_factor of 4, and a prefix distribution of $[[5, 10], [10, 10]]$. Here, a total of 20 text segments were sampled, with 10 having a prefix length of 5 and the other 10 a prefix length of 10. The covariance matrix is estimated from samples drawn from the "bigcode/the-stack[13]" dataset. The intervention is applied at MLP layer 19 of CodeLlama.

**MEMIT, PMET** Following PMET (Li et al., 2024b) and MEMIT (Meng et al., 2023), we perform a grid search over key hyperparameters to identify the configuration for code-related knowledge editing tasks. The hyperparameter sweep included: learning rates $\{0.5, 0.1, 0.05, 0.01, 0.005, 0.001\}$, mom2_update_weight $\{100, 500, 800, 10000, 15000, 20000\}$, v_weight_decay $\{0.5, 0.1, 0.05, 0.01, 0.005\}$, and clamp_norm_factor $\{0.5, 0.75, 1, 2, 5, 10, 15\}$. In addition, the covariance matrices are estimated from the "bigcode/the-stack" dataset, with mom2_n_samples of 100,000. Following the hyperparameter sweep, both PMET and MEMIT adopt the same final configuration: a learning rate of 0.01, a mom2_update_weight of 500, a v_weight_decay of 0.05, and a clamp_norm_factor of 2. (The experimental setup for PMET, as reported by Li et al. (2024b), closely follows the configurations employed for MEMIT.)

---

[12]https://github.com/kmeng01/rome/tree/main/hparams/ROME
[13]https://huggingface.co/datasets/bigcode/the-stack-v2-dedup

**Few-shot** Few-shot (Parnami & Lee, 2022) constructs prompts by directly providing the error message along with the Java-corrected D function, enabling the model to perform targeted code correction with minimal additional context.

**FiNE** Following defalt experimental setup (Pan et al., 2025), we freeze the final three layers and update the preceding layers. A learning rate of 0.001, determined via hyperparameter sweep, is applied, while all other hyperparameters remain consistent with the original FiNE configuration.

**LoRA** We target the q_proj and v_proj modules with a rank of 8 and a learning rate of 0.0001, restricting edits to a Transformer layer 19. All other model parameters remain frozen during training, following the default experimental setup as described in the baseline configuration.

**WISE** We follow the default experimental setup for WISE (Parnami & Lee, 2022), restricting updates to the designated inner parameters. Training is performed with a learning rate of 1.0 under an $L_\infty$-norm constraint of 1.0, while all other hyperparameters remain consistent with the original configuration[14].

**AGRACE** We follow the default experimental setup of AGRACE under CodeLlama (Li et al., 2025), restricting updates to the designated inner parameters and maintaining all other hyperparameters as in the original configuration.

**FT** We restrict parameter updates to MLP layer 19, which enables efficient knowledge modification while preserving the model's overall behavior. We train with a learning rate of 0.005 and impose an $L_\infty$ norm constraint (Zhu et al., 2020a) of 0.1, applying early stopping once the loss falls below 0.01 to prevent overfitting.

**TCPE** In the TCPE method, the hyperparameters for LTC4 and LTC8 are configured as follows: learning rate $\{0.05, 0.005\}$, mom2_n_samples of 500,000, clamp_norm_factor $\{13, 18\}$, and neuron activation thresholds $\tau \in \{0, 0.001, 0.01, 0.05, 0.08, 0.1, 0.15, 0.2\}$. Knowledge injection is applied at layer 19 of the TransCoder module. In this process, the covariance matrix samples from the "bigcode/the-stack" dataset. In addition, to construct the random prefix distribution, we use a single prefix of length 10. To prevent overfitting, an early stopping mechanism is employed during training. If the negative log-likelihood loss shows no significant improvement over three consecutive steps, training is halted. The tolerance is set to 0.001.

### K.2 HYPERPARAMETER SETTINGS FOR NLP TASK

For experiments on knowledge editing in natural language tasks using Llama2, the baseline methods ROME, MEMIT, and PMET adhere to the hyperparameter settings reported by Pan et al. (2025). In the TCPE method, the hyperparameters for the LTC4 variant are set as follows: a learning rate of 0.5, mom2_n_samples of 500,000, clamp_norm_factor of 35, and neuron activation thresholds of $1 * 10^{-4}$. Knowledge injection is applied at layer 19 of the TransCoder module, with the covariance matrix estimated from samples of the Wikipedia dataset. To prevent overfitting, an early stopping mechanism is employed: if the negative log-likelihood loss does not improve significantly over three consecutive steps, training is halted, with a tolerance of 0.001.

## L TRAINING DETAILS FOR TRANSCODER MODULES

The training of TransCoder modules is guided by a loss function designed to balance faithful approximation of the original MLP outputs with sparsity of the internal feature representations. Formally, the loss function is defined as:

$$\mathcal{L}_{\text{TC}} = \underbrace{\left\| \text{MLP}^{(l)}(\bar{h}_i^{(l)}) - \text{TC}^{(l)}(\bar{h}_i^{(l)}) \right\|_2^2}_{\text{faithfullness loss}} + \lambda \underbrace{\left\| z_{\text{TC}}^{(l)}(\bar{h}_i^{(l)}) \right\|_1}_{\text{sparsity loss}} \tag{8}$$

The quality and effectiveness of the TransCoder modules mainly depend on two key hyperparameters: the learning rate and the $L_1$ regularization coefficient $\lambda$. This coefficient is balancing faithful

---

[14]https://github.com/opanhw/FiNE/tree/main/hparams/LoRA

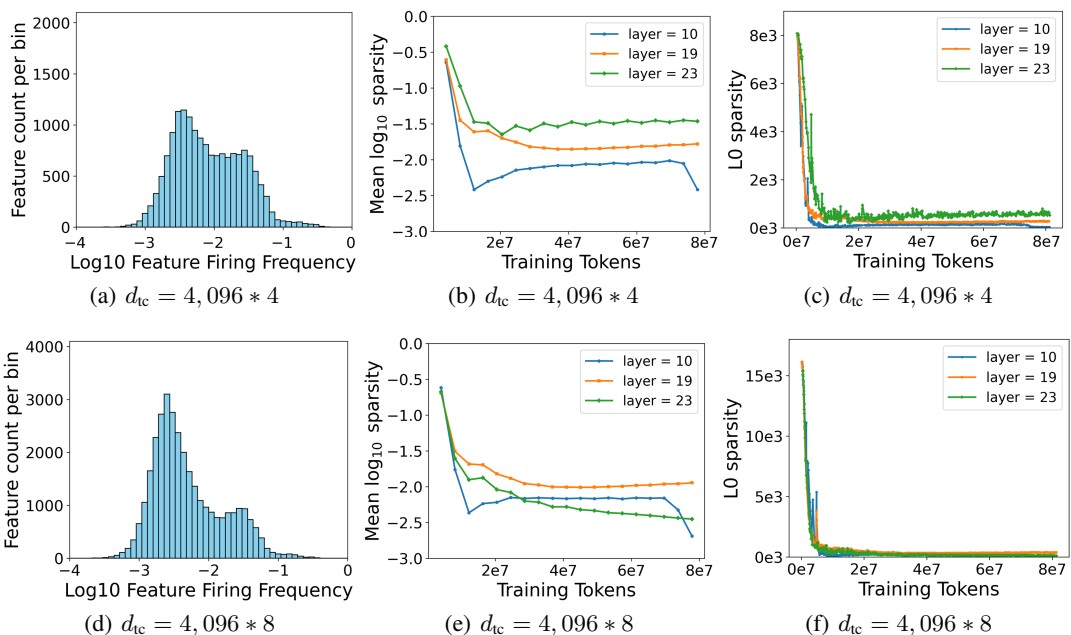

Figure 10: **Comparison of sparsity characteristics for TransCoder modules with $d_{\text{tc}} = 4,096 * 4$ and $d_{\text{tc}} = 4,096 * 8$.** Figures 10(a) and 10(d) show the $log_{10}$ firing frequency distribution of features at the end of training for a TransCoder module at layer 19. Figures 10(b) and 10(e) report the $L_0$ sparsity, i.e., the number of active features per input token for layers 10, 19, and 23. Figures 10(c) and 10(f) track the mean $log_{10}$ activation sparsity throughout training for the same layers.

approximation of the original MLP's outputs against maintaining a high degree of sparsity on its internal feature space $z_{\text{TC}}^{(l)}$ [15].

## L.1 EXPERIMENTAL CONFIGURATION

**Datasets.** The TransCoder is trained on intermediate activations produced by the target LLM during inference, without requiring access to the original pretraining data. As long as the input text can trigger the target MLP's computations, these activations can be collected for training. Such input data can be any publicly available text, eliminating the need to access the pretraining dataset. For CodeLlama, the TransCoder modules were trained on 80 million tokens sampled from the `"codeparrot/github`[16]`"` dataset. For Llama2, the TransCoder modules were trained on 10 million tokens sampled from the `"Skylion007/openwebtext`[17]`"` dataset.

**Hyperparameters.** TransCoders were trained with learning rates ranging from $5 * 10^{-4}$ to $1 * 10^{-4}$, and the $L_1$ regularization coefficient $\lambda$ was varied between $1 * 10^{-8}$ and $1 * 10^{-6}$. This range consistently produced robust results across diverse experimental settings. Regularization values above this range tended to induce excessive sparsity, leading to a substantial number of inactive features that failed to capture meaningful information. Conversely, sparsity constraints below this range resulted in many features activating frequently and indiscriminately across contexts.

All TransCoders were initially trained to replace the MLP layer 19 of the target model architecture. Once a viable hyperparameter configuration was identified for a given expansion factor, the same configuration was reused to train TransCoders for additional layers in order to evaluate whether our approach generalizes across the model. Here, hyperparameters selected for layer 19 were reused for layers 10 and 23. After training, we assess the quality of a TransCoder by examining its sparsity behavior. We present the distribution of feature firing frequencies (Figures 10(a) and 10(d)) for LTC4 and LTC8. A well-formed distribution appears smooth on a log scale and avoids heavy tails, i.e., few features should be consistently inactive or always firing, both of which indicate poor utilization. For

---

[15]To this end, $d_{\text{tc}}$ is typically chosen multiple times larger than $d_{\text{mlp}}$

[16]https://huggingface.co/datasets/codeparrot/github-code

[17]https://huggingface.co/datasets/Skylion007/openwebtext

layer 10, feature sparsity increased toward the end of training, with more inactive features observed in the $\log_{10}$ firing frequency. This effect aligns with prior findings: early-layer transcoders often show lower quality (Dunefsky et al., 2024), and editing in early layers tends to be less effective (Meng et al., 2022).

## L.2 TRAINING TIME

Existing knowledge editing methods, such as ROME, MEMIT, PMET, WISE, and AGRACE, require significant computational resources and **hours-to-days** to compute covariance matrices or build external memory modules. Notably, this cost is one-time and reusable for multiple edits.

Similarly, in our study, we trained TransCoders on an internal cluster with NVIDIA H100 PCIe GPUs (80GB VRAM). For codeLlama-7b-Instruct, training with an intermediate dimension of $4,096 * 4$ took about 2 hours and 20 minutes on a single GPU. Notably, TransCoder supports **one-time training** that can be **reused** across all locate-and-edit methods for interpretability analyses. It provides intuitive visualizations that reveal the correspondence between edited neurons and injected knowledge, enabling developers to transparently track the knowledge injection process.

## L.3 TRANSCODER ADAPTER: FAST INTEGRATION OF THE TRANSCODER MODULE

We design a Transcoder Adapter that wraps TransCoder as a standard PyTorch submodule. This adapter serves as a plug-and-play replacement for MLP layers: TransCoder parameters are loaded from its pretraining, while all other model weights remain unchanged from the original pretrained checkpoint. Here, the loading process takes only a few seconds. This approach is similar in spirit to the integration of LoRA modules. As illustrated in Figure 11, we provide an example of loading and integrating the TransCoder module.

We ran multiple experiments on the G4GD dataset using the same device to compare the total runtime of ROME across different architectures (MLP, LTC4, LTC8). The results show that ROME's runtime is $17\,\mathrm{m}\,9\,\mathrm{s} \sim 17\,\mathrm{m}\,45\,\mathrm{s}$ on MLP, $17\,\mathrm{m}\,15\,\mathrm{s} \sim 18\,\mathrm{m}\,6\,\mathrm{s}$ on LTC4, and $17\,\mathrm{m}\,2\,\mathrm{s} \sim 17\,\mathrm{m}\,52\,\mathrm{s}$ on LTC8. The slight variation in runtime arises from differences in model architecture and size. Empirical evaluation on 600 samples demonstrates that integrating the TransCoder module introduces additional inference latency, resulting in a total increase of several seconds.

```python
def load_model_and_tokenizer(args):
    model = AutoModelForCausalLM.from_pretrained(MODEL_path)

    if load_Transcoder:
        adapter = TranscoderAdapter.load(Transcoder_path)
        model.base_model.layers[layer_num].mlp = adapter

    tok = AutoTokenizer.from_pretrained(MODEL_path)
    return model, tok
```

Figure 11: **Efficient Layer Replacement in Transformers via TransCoder Adapters.**

