# OpenReview forum: "Precise and Interpretable Editing of Code Knowledge in Large Language Models"
_ICLR.cc/2026/Conference — ICLR 2026 Poster_

### Official Review · Reviewer_FKNG · 2025-10-26

**Soundness:** 3
**Presentation:** 3
**Contribution:** 2
**Rating:** 4
**Confidence:** 4

**Summary:**

This paper proposes the TCPE model editing method based on TransCoder. This method replaces the original MLP layers with an MLP-like model component that has a wide and sparsely activated hidden feature vector, avoiding the imprecision and uninterpretable edits caused by the neuronal polysemanticity of the original MLP. The paper also introduces KECode, a new code-to-code model editing benchmark. Experiments on KECode and existing model editing benchmarks demonstrate the effectiveness of TCPE.

**Strengths:**

* Replacing the original MLP with a wider and sparse MLP using TransCoder is novel and interesting.
* KECode, a code-to-code model editing benchmark, is proposed, providing a valuable resource for the community.
* The effectiveness of TCPE is demonstrated on the KECode, ZsRE, and CounterFact datasets.
* An in-depth analysis of TransCoder neurons is conducted, offering valuable insights.

**Weaknesses:**

* TCPE lacks principled innovation, as both TransCoder and ROME-like editing are existing works.
* The evaluation of the model editing task seems limited to single-case edits, lacking assessments closer to real-world scenarios such as sequential or batch edits.
* There is no discussion of TCPE’s scalability; its performance when editing large batches or performing sequential edits remains unknown.
* TransCoders require additional training, and the training environment appears to be more demanding compared to the baselines. Moreover, it is unclear whether replacing the MLP with TransCoders affects the model’s original capabilities.

**Questions:**

Would the introduction of TransCoders increase inference latency?

---

> ### Author Response · Authors · 2025-11-14
> **Official Comment by Authors (Part_1)**
>
> We sincerely appreciate your valuable feedback and guidance on potential directions for future research. We would be very glad to provide further clarification on the technical details and other aspects of our work.
>
> ### **Weakness 1:**
>
> We fully understand and appreciate your concerns. Our work does not merely combine ROME and TransCoder. Instead, we establish a robust bridge between the two through continuous validation and analysis from the perspectives of interpretability, localizability, and controllability. This process is structured around the following research questions:
>
> **RQ1: How does TransCoder affect loading efficiency, inference speed, and the original model performance?** We first evaluate its loading and inference efficiency, and analyze its impact on the original model’s performance (**see Section 4.3 and  Appendix L.3**).
> - **RQ1.1 Loading and inference efficiency.** As described in Appendix L.3, we design a **Transcoder Adapter** that wraps TransCoder as a standard PyTorch submodule. This adapter serves as a plug-and-play replacement for MLP layers: TransCoder parameters are loaded from its pretraining, while all other model weights remain unchanged from the original pretrained checkpoint. The loading process takes only a few seconds and has negligible impact on inference latency. We ran multiple experiments on the G4GD dataset using the same device to compare the total runtime of ROME across different architectures (MLP, LTC4, LTC8). The results show that ROME’s runtime is 17 m 9 s \~ 17 m 45 s on MLP, 17 m 15 s \~ 18 m 6 s on LTC4, and 17 m 2 s \~ 17 m 52 s on LTC8. Due to stochasticity in generative outputs, runtimes vary slightly, but differences across architectures are minimal, indicating that incorporating TransCoder does not significantly affect overall latency.
> - **RQ1.2 Preservation of original model  performance.** In Table 5, we systematically evaluate whether replacing MLP layers with TransCoder modules of varying sizes degrades the original model performance. The results show that across layers and module sizes, the model’s original performance remains largely unchanged.
>
> **RQ2: Can we precisely identify knowledge-carrying neurons in sparse TransCoder?** The sparse activation patterns of TransCoder offer a potential advantage for knowledge localization, and we systematically verify its localizability and separability (**see Appendix G**).
> - **RQ 2.1: Consistency of Activation Patterns for the Same Error Type.** In Table 11, we observe that for the same type of error, the active neurons in TransCoder largely overlap. This high consistency suggests that  target error-related knowledge is represented at fixed neuron positions, which enables reliable knowledge localization.
> - **RQ 2.2: Separability of Activation Subspaces Across Different Error Types.** In Table 10, we further observe  that the active neurons for different error types exhibit minimal overlap. This indicates that TransCoder structurally decouples semantic representations, enabling precise and localized editing of different types of errors.
>
> **RQ3: Does the sparsity and monosemanticity of TransCoder improve interpretability in knowledge editing?** We further evaluate whether TransCoder’s higher monosemanticity leads to improved interpretability in editing (**see Section 4.2**).
> - **RQ3.1 Blind interpretability comparison.** In Section 4.2 (Part 2), we conduct a *Blind Interpretability Comparison*, showing that TransCoder neurons are significantly more interpretable.
> - **RQ3.2 Visualization of high-activation neuron patterns.** In Section 4.2 (Part 1), we visualize examples triggering high-activation neurons (e.g., errors related to `str.length`) and observe that TransCoder neurons exhibit highly concentrated, semantically consistent activations. This indicates that these highly active neurons possess semantic specificity and are directly correlated with the target knowledge, thereby providing interpretability for knowledge editing.
>
> **RQ4: How do neurons with different activation levels contribute to editing effectiveness?** We investigate the role of neurons with varying activation levels during knowledge injection (**see Section 4.3**).
>
> - **RQ4.1 Contribution of high-activation neurons.** Using threshold-controlled experiments(Tables 3), we update high-activation neurons based on different activation levels. Results show that updating fewer than 1% of the highly active neurons is sufficient for effective knowledge injection, while medium-activation neurons have limited impact.This indicates that knowledge in TransCoder is highly concentrated, with a few highly active neurons carrying the majority of semantic information.
>
> - **RQ4.2 Contribution of low-activation neurons.** Further ablation experiments (Tables 4) show that updating only low- or inactive neurons almost never achieves effective knowledge injection or error correction, confirming the central role of high-activation neurons in editing.

---

> ### Author Response · Authors · 2025-11-14
> **Official Comment by Authors (Part_2)**
>
> ### **Weaknesses 2 and 3:**
> - We greatly appreciate your valuable suggestions for future improvements to our work. In the revised manuscript we added the Appendix A “Limitations and Broader Impact” which discusses these aspects as potential directions for future research. As you noted, the purpose of TCPE is to serve as a tool for understanding the underlying principles of knowledge editing: it focuses on interpreting the relationship between limited injected knowledge and its corresponding edits, and is not intended as a practical method for large-scale knowledge updates. Exploring TCPE’s interpretability in scenarios such as batch or sequential edits will be a focus of our future research.
>
> ### **Weakness 4:**
>
> We sincerely thank you for your valuable suggestions and fully understand your concern. Considering that the current locate-and-then methods primarily focus on improving editing performance while lacking an understanding of the editing sites and injected knowledge, providing interpretability is indeed necessary.
>
> -  In the field of Transformer interpretability, methods designed to enhance interpretability inevitably incur additional computational overhead, due to the training of auxiliary interpretability modules such as TransCoder or sparse autoencoders. However, **these trainings are one-time procedures and can be reused**.  Moreover, **the training barrier for TransCoder is low**. It is trained on intermediate activations produced by the target LLM during inference, without accessing the original pretraining dataset. As long as the input text can trigger the target MLP’s computations, these activations can be collected for training, and the input data can be any publicly available text.  We will also pay attention to emerging lightweight methods in interpretability for future work.
> - In Section 4.3 (Table 5), we systematically evaluate whether replacing MLP layers with TransCoder modules of varying sizes degrades the original model performance.  As shown in Table 5, model performance remains stable across different TransCoder widths and positions on the G4GD dataset.  The accuracy changes are marginal, with some configurations even slightly outperforming the original MLP, indicating that integrating TransCoder has negligible effect on the model's overall performance.
> - In addition to performance metrics, we also analyze the error patterns across different model variants. As shown in Appendix Table 14, the distribution of top error clusters remains largely unchanged after replacing the MLP with TransCoder modules. This indicates that the core behavioral characteristics of the model are preserved.  These results demonstrate the functional compatibility of the TransCoder module with standard MLP layers. It not only preserves predictive performance but also retains error-specific activation patterns, making it a reliable substitute for evaluating and manipulating the model's internal knowledge.
>
> ### **Question:**
> - As described in Appendix L.3, we design a **Transcoder Adapter** that wraps TransCoder as a standard PyTorch submodule. This adapter serves as a plug-and-play replacement for MLP layers: TransCoder parameters are loaded from its pretraining, while all other model weights remain unchanged from the original pretrained checkpoint. The loading process takes only a few seconds and has negligible impact on inference latency. We ran multiple experiments on the G4GD dataset using the same device to compare the total runtime of ROME across different architectures (MLP, LTC4, LTC8). The results show that ROME’s runtime is 17 m 9 s \~ 17 m 45 s on MLP, 17 m 15 s \~ 18 m 6 s on LTC4, and 17 m 2 s \~ 17 m 52 s on LTC8. Due to stochasticity in generative outputs, runtimes vary slightly, but differences across architectures are minimal, indicating that incorporating TransCoder does not significantly affect overall latency.
> - This approach is conceptually similar to the integration of LoRA modules, as both methods introduce auxiliary components while keeping the original model weights largely unchanged. The following example illustrates how to use the TransCoder Adapter to load and integrate the TransCoder module:
>     ```python
> def load_model_and_tokenizer(args):
>         model = AutoModelForCausalLM.from_pretrained(MODEL_path)
>         if load_Transcoder:
>             adapter = TranscoderAdapter.load(Transcoder_path)
>             model.base_model.layers[layer_num].mlp = adapter
>         tok = AutoTokenizer.from_pretrained(MODEL_path)
>         return model, tok
>     ```

---

> > ### Comment · Reviewer_FKNG · 2025-11-17
> >
> > The statement “Due to stochasticity in generative outputs, runtimes vary slightly” is incorrect. The inference time of an LLM is influenced by factors such as the model architecture and size, the computational capability of the hardware, the inference framework (e.g., FP16, quantization), batch size, parallelism, and hardware load—not by so-called “stochasticity in generative outputs.” The results reported by the authors indeed show that the Transcoder Adapter introduces additional inference latency, and this should be clearly stated in the paper.

---

> ### Author Response · Authors · 2025-11-17
>
> We sincerely appreciate your clarification and fully agree with your suggestion. You are absolutely right that factors such as model architecture, hardware performance, numerical precision, and runtime load are the primary determinants of inference latency.
>
> Our previous statement referred to variation in end-to-end wall-clock runtime, which fluctuates due to sampling-induced differences in output length, particularly in code tasks. We apologize for the imprecise wording.
>
> Following your suggestion, we have revised the inaccurate statements in the paper. The updated version can be found in Appendix L.3.
>
>  > The variation in runtime arises from differences in model architecture and size. Empirical evaluation on 600 samples demonstrates that integrating the TransCoder module introduces additional inference latency, resulting in a total increase of several seconds.

---

> ### Author Response · Authors · 2025-11-26
>
> Dear Reviewer FKNG,\
> Thank you for your feedback! As requested, we have revised the manuscript accordingly. To quantify the computational overhead, we conducted an empirical latency evaluation on 600 randomly selected samples. The results show that integrating the TransCoder module adds only a marginal cost, an increase of less than **0.05 seconds per sample** on average. The total additional latency amounts to only a few seconds across the entire evaluation. This effect is consistent across different prompts, indicating that the overhead introduced by TransCoder is systematically negligible in practical use.
>
> **If you have any additional questions or suggestions, we would be more than happy to provide further clarification. We hope these revisions address your concerns, and if they meet your expectations, we would greatly appreciate your favorable consideration in scoring.**
>
> Best regards,\
> Authors

---

> ### Author Response · Authors · 2025-11-27
>
> Dear Reviewer FKNG,\
> Thank you very much for your positive evaluation and for raising the score. We sincerely appreciate your expert guidance and valuable feedback on our work. Your insights are immensely helpful. Wishing you all the best in your research endeavors!
>
> Best regards,\
> Authors

---

### Official Review · Reviewer_sF6r · 2025-10-30

**Soundness:** 3
**Presentation:** 3
**Contribution:** 3
**Rating:** 8
**Confidence:** 3

**Summary:**

This paper makes two key contributions to improve the precision and interpretability of knowledge editing in code LLMs on code translation domain.

First, it proposes TransCoder-based Precise Editing, which is a method that replaces the Transformer’s MLP layer with a sparse TransCoder module, enabling edits that target specific highly activated neurons associated with a particular piece of code knowledge. This design allows localized and interpretable edits, minimizing side effects on unrelated behavior while providing clear neuron-level insight into how knowledge is inserted.

Second, the authors introduce KECode, a new benchmark for evaluating knowledge editing in code translation. KECode consists of 600 Java-to-D translation examples paired with unit tests for functional correctness verification. Using KECode, the paper shows that TCPE significantly outperforms existing methods such as ROME, MEMIT, and PMET on correcting Java-to-D translation errors.

**Strengths:**

* For originality, TCPE presents a novel mechanism that leverages the sparsity and monosemanticity of TransCoder neurons to perform more precise, localized edits.
* For clarity, the authors place strong emphasis on interpretability. Because TCPE operates at the neuron level, they can explicitly identify which neurons are edited and link them to corresponding knowledge changes. The empirical finding that “highly active neurons carry more essential information during knowledge injection” is well-supported and insightful, strengthening the paper’s interpretability claims.
* For quality, the evaluation is methodologically solid. The paper reports multiple complementary metrics, including efficacy, specificity, and reliability, as well as detailed ablation studies and granular analysis tailored to the knowledge editing context.

**Weaknesses:**

* For scope and generalization, this paper focuses solely on Java-to-D code translation. It is unclear whether the approach generalizes to other software engineering tasks such as code completion, bug fixing, or program repair. The KECode benchmark evaluates functional error correction, which is one specific type of knowledge editing. It would also be interesting to discuss or demonstrate applicability to, for example, inserting new API knowledge or modifying non-functional aspects of code, which surface broader SWE contexts.
* For practicality and Integration, TCPE requires modifying the model architecture by replacing MLP layers with TransCoder modules. The paper does not clarify whether this change requires retraining the new layers or fine-tuning the entire model. Additional discussion on how TCPE integrates with other architectures (e.g., MoE) would help assess its practical adoption potential.
* For baseline adaptation, while TCPE outperforms existing NLP-based editing methods, like ROME, MEMIT, etc, these baselines were originally designed for factual knowledge editing in natural language models, not code. Because code has strict syntax and execution semantics, these baselines may be disadvantaged. The paper would benefit from either a stronger code-specific baseline, or a clearer discussion of how baseline implementations were adapted to ensure fairness.

**Questions:**

Two questions:

* How were the specific layers {10, 19, 23} selected for replacing MLP layers with TransCoder modules? Were these layers empirically identified or based on prior interpretability insights about CodeLlama?
* When initializing a TransCoder-modified model from pretrained weights, how are the parameters loaded or transferred to the new architecture? Is there a compatibility or adaptation step between the original MLP weights and the TransCoder modules?

---

> ### Author Response · Authors · 2025-11-14
> **Official Comment by Authors (Part_1)**
>
> We sincerely appreciate your positive feedback on our work, and we are also very grateful for the improvement directions you suggested. For each of the points you raised, we have examined the current challenges and potential solutions.
>
> **Weakness 1:**
> - We are grateful for your constructive feedback on enhancing our method. We fully agree with your suggestions and would like to share our thoughts. Our current study indeed focuses on the code translation task and employs the KECode benchmark for evaluation. This choice is primarily motivated by the fact that code translation and functional error correction are relatively structured and easily verifiable knowledge editing tasks, where the input (original code) and the expected output (corrected code) have a clear correspondence. Such characteristics provide a solid foundation for method development and evaluation. Nevertheless, validating knowledge editing in broader software engineering tasks, such as code completion or automated program repair, remains an acknowledged limitation.
>
> - We fully agree on the importance of extending the knowledge editing method to tasks such as "inserting new API knowledge" or "modifying non-functional code properties", and we have summarized several potential technical challenges in doing so. First, translating non-functional requirements into precise, model-executable code changes is itself a non-trivial problem. Inserting new API calls or modifying architectural properties often requires understanding a broader code context, potentially spanning the entire project, and localized edits may have unintended side effects on other modules. Second, existing evaluation metrics are insufficient for these tasks, necessitating the development of new quantitative measures to assess editing effectiveness.
>
> - To address these challenges, we plan to explore several directions in future work to validate and extend the applicability of our method. First, we aim to design fine-grained instruction templates or constraints tailored for more complex editing tasks, helping the model clearly understand the objectives and boundaries of non-functional modifications, thereby mitigating potential side effects. Second, we plan to investigate hierarchical editing or multi-turn interaction strategies, enabling the model to comprehend code context at multiple levels, interact with developers for confirmation, or execute complex refactoring, performance optimization, or architectural modification tasks step by step. This approach is expected to handle edits spanning multiple modules or cross-function dependencies. Finally, we will explore the development of new benchmarks and quantitative metrics to systematically evaluate the effectiveness, reliability, and overall impact of non-functional code modifications. Through these efforts, we hope to assess the feasibility and effectiveness of our method across a wider range of software engineering scenarios.
>
> **Weakness 2:**
> - In TCPE, it is not necessary to retrain the entire model; only the target layers need to be retrained. Specifically, each TransCoder module is trained using the activations of the original MLP, allowing it to replicate the input-output mapping while providing a wider internal feature dimension. Importantly, each module preserves the same input and output hidden dimensions as the original MLP, ensuring seamless replacement within the Transformer layers. Additional training details are provided in Appendix L.
> - We greatly appreciate your insightful comments and fully recognize that evaluating TCPE within MoE architectures is a valuable and essential direction for future research. Although our current work focuses on Transformers with dense MLP architectures, we are actively investigating how to extend TCPE to MoE. Our method reveals a direct correspondence between injected knowledge and edited neurons in TransCoder, which suggests that MoE’s sparse activation mechanism may offer a cleaner and more precise pathway for neuron-level knowledge editing. In future work, we plan to validate the applicability of TCPE to MoE architectures in the knowledge editing field, focusing on aspects including: (i) Whether neurons within each expert exhibit fact-specific activations similar to those observed in dense TransCoders.  (ii) How to balance updates across multiple experts when the knowledge spans more than one routing path.  (iii)  Whether MoE’s gating mechanisms can consistently route knowledge-relevant inputs to the same expert for effective editing. (iv) Given the sparsity of MoE, experts may contain under-utilized neurons. Whether forcing updates on these "inactive" neurons can complement the editing process without perturbing high-activity neurons.

---

> ### Author Response · Authors · 2025-11-14
> **Official Comment by Authors (Part_2)**
>
> **Weakness 3:**
>
> - We fully understand your concern. In our evaluation, we considered three representative baselines for code-related knowledge editing: AGRACE, few-shot prompting, and fine-tuning. AGRACE is the most recent method specifically designed for knowledge editing in code LLMs. Few-shot prompting, where a small set of examples is provided at inference time to guide model behaviour, is a common strategy in code-related tasks. Fine-tuning (FT) is also a widely adopted adaptation strategy in code-related LLM research.
>
>     **Table 1. Comparative Performance of Knowledge Editing Methods in a Multi-Error Editing Scenario.** Score quantifies overall knowledge editing performance and is calculated as Score = (GN − CD) + (LoC − DT) + RE.
>   | Method | Score ↑ | GN ↑ | CD ↓ | LoC ↑ | DT ↓ | RE ↑ | AC_post | AC_pre |
>   |-----------------|-----------|-----------|-----------|-----------|-----------|-----------|-----------|-----------|
>   | FT           | 86.40  | 7.09  | 100.00 | 87.36 | 6.38 | 97.97 | 56.33 | 57.50 |
>   | Few-shot     | 120.20 | 12.06 | 87.23  | 90.20 | 2.61 | 107.78 | 61.33 | 57.50 |
>   | AGRACE       | 99.85  | 0.71  | 100.71 | 99.56 | 0.00 | 100.29 | 57.67 | 57.50 |
>   | TCPE (LTC4)  | 171.45 | 32.86 | 46.43  | 82.17 | 6.86 | 109.71 | 64.00 | 58.33 |
>
> - As shown in the Table 1 above, TCPE significantly improves both generalization and specificity compared to all baseline methods. It also achieves the highest reliability and post-editing accuracy, demonstrating its robustness and effectiveness in the code-related task. AGRACE, a recent code-focused knowledge editing method, has been evaluated using text-based metrics [2]. In our work, we further employ functionally equivalent metrics to assess AGRACE, which are more standard and reliable for evaluating the quality of generated code in the code domain. When applying AGRACE to KECode, we observe a substantial drop in performance. As reported in Table VI of the original paper [2], the post-editing specificity is even lower than that of the pre-edited model, indicating that AGRACE struggles to prevent unintended interference with non-target knowledge. These observations are consistent with our findings.
>
> **Question 1:**
>
> - For code-related tasks, as described in Appendix H, we introduce a fine-grained causal intervention method for token-by-token analysis, specifically designed to identify where knowledge is stored in the Code LLM. By systematically corrupting the single token in the input sequence and examining the layer at which an intervention on the corresponding hidden states most effectively restores the correct output, we find that code-related knowledge (*subject* $s$) for a target token ($o$) is predominantly stored in the middle-to-late layers. Furthermore, Voita et al. (2019) proposed that representations in the deeper layers of a Transformer predominantly encode aggregated contextual and task-relevant information for output generation, rather than maintaining discrete, token-level knowledge[1]. Therefore, we choose the middle layers for code knowledge injection.
>
> **Question 2:**
>
> - The Transcoder module is trained on the activations of the original MLP, allowing it to mimic the input-output mapping of the MLP while providing a wider internal feature dimension. Importantly, the module maintains the same input and output hidden dimensions as the original MLP, ensuring seamless replacement within a Transformer layer. In other words, its external interface remains fully compatible with the pretrained model.  When initializing a Transcoder-modified model from pretrained weights, all parameters of the base language model are loaded directly from the original checkpoint without any modification. The only architectural change occurs at the target layer, which loads the Transcoder module instead of the original MLP.
> - We design a **Transcoder Adapter** that wraps the Transcoder as a standard PyTorch submodule. This Transcoder Adapter provides a plug-and-play replacement for MLP layers: the Transcoder module comes from Transcoder pretraining, while all other model weights are loaded from the original pretrained checkpoint. The loading process is very fast, taking just a few seconds. Below is a usage example of the adapter:
>     ```python
> def load_model_and_tokenizer(args):
>         model = AutoModelForCausalLM.from_pretrained(MODEL_path)
>         if load_Transcoder:
>             adapter = TranscoderAdapter.load(Transcoder_path)
>             model.base_model.layers[layer_num].mlp = adapter
>         tok = AutoTokenizer.from_pretrained(MODEL_path)
>         return model, tok
>     ```
>
> [1] Voita, E., Sennrich, R., & Titov, I. (2019). The Bottom-up Evolution of Representations in the Transformer: A Study with Machine Translation and Language Modeling Objectives. In EMNLP 2019.
>
> [2] Li X, Wang S, Li S, et al. (2025). Model Editing for LLMs4Code: How Far are We? In ICSE 2025.

---

### Official Review · Reviewer_VVHx · 2025-10-31

**Soundness:** 3
**Presentation:** 2
**Contribution:** 2
**Rating:** 6
**Confidence:** 1

**Summary:**

This paper addresses the challenge of precise and interpretable knowledge editing (KE) in Large Language Models (LLMs) for code-related tasks. The authors argue that existing KE methods, which often target standard MLP layers, are hindered by neuronal polysemanticity, leading to imprecise edits and poor interpretability. To solve this, the authors propose TransCoder-based Precise Editing (TCPE). This method involves two key stages:
1. Architectural Modification: The standard MLP layer in a target Transformer (CodeLlama-7b-Instruct) is replaced by a "TransCoder" module—a sparse, wide, MLP-like component from prior work (Dunefsky et al., 2024) that is trained to have more monosemantic neurons.
2. Editing Mechanism: A ROME-like update is applied, but it is restricted only to the small set of "active neurons" in the TransCoder module that are relevant to the knowledge being corrected.

For evaluation, the authors introduce KECode, a new benchmark for code-to-code translation (Java-to-D) that uses functional equivalence (i.e., unit test pass/fail) as the success metric. Their experiments show that TCPE on the modified architecture (e.g., "LTC4") outperforms baseline KE methods (ROME, MEMIT, etc.)

**Strengths:**

Knowledge editing is an important research direction to save computational resources by avoiding retraining.

**Weaknesses:**

I'm not familiar with this field, so I will give my confidence score to 1. Please lower my score weight for this paper.

**Questions:**

Does the method apply to more applications and datasets?

---

> ### Author Response · Authors · 2025-11-14
>
> We sincerely appreciate your feedback on our work. Considering that our research directions may not completely align, we would like to briefly introduce the motivation behind our study and address the questions you raised. If you have any other questions of interest, we would be happy to provide further clarification.
>
> **Motivation:**
> - Existing locate-and-edit methods are primarily built on the ROME approach. Although these methods improve editing performance, their interpretability remains limited, making it difficult to understand how knowledge is injected into the MLP layers. TransCoder, a recently introduced tool in the field of Transformer interpretability, offers a sparse and monosemantic activation space that features clear visualization of internal information flow and activation patterns.
>
> - Building on this approach, we propose TCPE, which establishes a bridge between ROME and TransCoder through systematic validation and analysis from the perspectives of interpretability, localizability, and controllability. This approach reveals a clear correspondence between the edited neurons and the injected knowledge, thereby laying the foundation for interpreting locate-and-edit methods.
>
> - One advantage of TCPE is that provides intuitive visualization (see Section 4.2 and Appendix F) which enables developers to transparently track the process and location of knowledge injection. Furthermore, TCPE leverages TransCoder’s sparse and monosemantic neurons, allowing it to modify only the neurons most relevant to the target knowledge and thereby greatly reducing editing side effects.  This improves over ROME-based methods which largely do not consider the polysemanticity of MLP neurons, causing direct edits to inadvertently affect non-target knowledge.
>
>
> **Question:**
>
> Thank you for your insightful comment and important question. In response, we first introduce the datasets used in this paper, followed by concrete interpretability visualizations and analyses of the overlap of active TransCoder neurons across and within error-type clusters.
>
> - Our evaluation extends beyond the G4GD dataset to include well-established benchmarks in code generation (HumanEval) and general NLP knowledge editing (CounterFact, zsRE). As reported in Appendices E.1 and E.3, TCPE demonstrates enhanced specificity.
>
> - In additioin, in Section 4.2, we visualize the input texts that high-activation neurons respond to, presenting direct evidence of the correspondence between these neurons and the injected knowledge. For example, focusing on a typical D-type conversion error `Error: cannot implicitly convert expression 'str.length' of type 'ulong' to 'int'`, we analyze the interpretability differences between active and inactive features from the LTC4 TransCoder module.  Figure 2 shows a representative example from the top-10 active features, which consistently respond to key tokens such as `str`, `string`, or `=.length`, directly related to the target error (More visualization results can be found in Appendix F.2 and Openreview Supplementary Material). In contrast, Appendix F.3 presents examples from 10 randomly inactivated features, which respond to structural or control-flow tokens like `if`, `N`, `ps`, and `;`. Although these inactive features also exhibit stable activation patterns, their captured knowledge is largely unrelated to the target error.
>
> - In Appendix G, we further analyse the overlap of active neurons across different error types in both MLP and TransCoder modules. TransCoder exhibits significantly lower cross-error overlap, indicating more specialized neurons for distinct error types. Furthermore, within clusters of the same error type, TransCoder neurons show higher intra-cluster overlap and more concentrated activation patterns compared to MLP neurons.

---

> ### Author Response · Authors · 2025-11-25
>
> Dear Reviewer VVHx,
>
> We sincerely appreciate the time and effort you have devoted to evaluating our work. We fully understand that unfamiliarity with the specific research domain may make it challenging to assess the contribution of our paper, and we truly appreciate your openness in expressing this concern.
>
> To the best of our knowledge, our work is the first to provide interpretability for locate-and-then methods. To facilitate your evaluation, we have provided substantial clarifications and supporting materials, including:
>
> - A full release of the dataset used in our experiments in the OpenReview supplementary material.
> - Detailed descriptions of the G4GD datasets in Section 3.2 and Appendix J (data sources, dataset sizes, comparisons with standard datasets, and prompt instructions).
> - A comprehensive list of all datasets, metrics, and baselines in Appendix D.
> - Reproducible training parameters in Appendices K and L.
> - We analyzed the inference overhead introduced by our framework modifications. The results show that the additional cost is minimal relative to the interpretability benefits (e.g., around 15 seconds for 600 samples).
>
> We are truly grateful for your time and appreciate your thoughtful consideration of our revised submission. **If the supporting evidence provided helps strengthen your confidence in the paper and lead to a more favorable assessment, we would be sincerely grateful! Should you have any further questions, we would be more than happy to address them!**
>
>
> Best regards,\
> Authors

---

### Official Review · Reviewer_76ke · 2025-10-31

**Soundness:** 2
**Presentation:** 1
**Contribution:** 2
**Rating:** 4
**Confidence:** 3

**Summary:**

This paper proposes a TransCoder-based Precise Editing (TCPE) method, to edit code knowledge in large language models. It also presents a new benchmark, KECode, for code-to-code translation based on functional equivalence. Experimental results
demonstrate that TCPE outperforms existing KE methods, achieving a substantial improvement of translation accuracy of CodeLlama-7b-Instruct from 57.5% to 64.0% in a low-resource scenario of Java-to-D translation.

**Strengths:**

1. This paper proposes a new knowledge editing method for code LLMs. There is also a new benchmark for evaluating the performance of LLMs on code-to-code translation.
2. Experimental results show that, TCPE outperforms existing knowledge editing methods with significant margins.
3. A neuron-level interpretability mechanism is introduced to effectively indicates the connection between the edited neurons and the inserted knowledge.

**Weaknesses:**

1. Neither codebase nor dataset is provided to confirm the reproducibility.
2. This paper lacks discussions of limitations and broader impact.
3. The presentation should be improved. For example, fonts in tables and figures can be larger for better reading experience.

**Questions:**

Would you like to enlarge the fonts in Figure 1, Figure 2, Table 3 and Table 4?

---

> ### Author Response · Authors · 2025-11-14
>
> We sincerely appreciate your valuable guidance on our paper. Following your suggestions, we have carefully revised the manuscript, which has substantially improved its clarity and readability. To facilitate your review, the changes have been highlighted in red. If our responses address your concerns, we would be deeply appreciative if you might consider adjusting your score. If you have any further questions or suggestions regarding our paper, we would be more than happy to address them.
>
> **Weakness 1:**
>
> Thank you for your feedback. The G4GD dataset was provided with the paper submission and can be found in the OpenReview supplementary material. Other publicly available datasets are described in Appendix D. Following your suggestion on reproducibility, we made the code available via [anonymous github link](https://github.com/Anonymous2025XX/AnonymouS). For the final version, we will further improve the readability of this code and remove irrelevant parts.
> Additionally, Section 3.2 and Appendix J provide detailed information on the G4GD datasets, including data sources, dataset sizes, comparisons with standard datasets, and prompt instructions. Appendix D lists all datasets, metrics, and baselines used, and Appendices K and L include the reusable training parameters. To improve readability, we have indicated the relevant appendix locations in the main paper.
>
> **Weakness 2:**
>
> Thank you for your valuable suggestions. We fully agree that a discussion of limitations and broader impact is essential for a high-quality paper. We have now added this section in the latest version, which can be found in Appendix A.
>
> **Weakness 3 and Question**
>
> We sincerely appreciate your feedback on the presentation. We have adjusted the font sizes of tables and figures to enhance readability and clarity.

---

> > ### Author Response · Authors · 2025-11-24
> >
> > Dear Reviewer 76ke,
> >
> > Thank you very much for your thoughtful and constructive feedback on our work. We have thoroughly revised the paper in accordance with all of your suggestions, and we sincerely believe that the overall quality has been improved under your guidance.
> >
> > In addition to the dataset originally included in the Supplementary Material and the additional datasets and training details provided in the Appendix during the initial submission in September, we have now released the core code of our paper as requested. We have also revised the figures and tables to address the previous font-size issues, and we have added a new Limitations and Broader Impact section.
> >
> > We greatly appreciate your time, effort, and guidance throughout this process, and we hope that the revisions adequately address your concerns and reflect meaningful improvements. If you have any further questions or require additional information, please do not hesitate to let us know, we would be more than happy to assist.
> >
> >
> > Sincerely,\
> > Authors

---

### Meta-Review · Area_Chair_Lgjn · 2026-01-13

**Summary:**

This paper proposes TransCoder-based Precise Editing (TCPE), a novel method for precise and interpretable knowledge editing in code large language models. By replacing standard MLP layers with novel modules which leverages the sparsity and monosemanticity of the TransCoder’s neurons for highly localized knowledge editing, TCPE exhibits neuron-level mechanistic interpretability characteristics that link edited neurons to specific code knowledge. The authors also introduce KECode, an equivalent benchmark for evaluating code-to-code translation editing. Based on this novel benchmark, the authors demonstrate that TCPE’s performance runs over nine baselines. Specifically, this method improves CodeLlama-7b-Instruct’s Java-to-D translation accuracy in low-resource scenarios.

**Reviewer Concerns:**

Strengths:

1. TCPE addresses critical limitations of existing knowledge editing methods by enhancing interpretability and reducing undesirable side effects through TransCoder’s structural advantages.

2. The proposed KECode benchmark bridges the gap in code-specific evaluation benchmarks. Also, the extensive experiments across four datasets validate the method's generalizability.

3. During the rebuttal period, the authors have addressed most of reviewer concerns, including releasing core code, adding a Limitations and Broader Impact section, and quantifying minimal inference latency overhead.

4. This method may be helpful to understand the behavior of LLMs in other domains.

Weaknesses:
Main limitations that reviewers are concerned about are as follows:

1. The method proposed in this paper focuses only on Java-to-D translation.

2. This paper lacks batch/sequential edit evaluations.

3. The comparison with other baselines is not fair enough.

Most of these limitations have been addressed during the rebuttal period through the authors’ adequate response.

**Reviewer Scores:**

The original scores are 8, 6, 4, 4, and the scores after rebuttal may be 8, 6, 6, 4. One reviewer decided to increase his/her rating. From my perspective, based on the significant innovations in interpretable code knowledge editing, the well-designed benchmark, and the authors’ comprehensive responses to reviewer feedback, the final result can be accept.

---

### Decision · Program_Chairs · 2026-01-26

Accept (Poster)